# Neural arbitration between social and individual learning systems

**Andreea Oliviana Diaconescu**[1,2,3,4†]*, **Madeline Stecy**[1,2,5†], **Lars Kasper**[1,2,6], **Christopher J Burke**[2], **Zoltan Nagy**[2], **Christoph Mathys**[1,7,8], **Philippe N Tobler**[2]

[1]Translational Neuromodeling Unit, Institute for Biomedical Engineering, University of Zurich & ETH Zurich, Zurich, Switzerland; [2]Laboratory for Social and Neural Systems Research, Department of Economics, University of Zurich, Zurich, Switzerland; [3]University of Basel, Department of Psychiatry (UPK), Basel, Switzerland; [4]Krembil Centre for Neuroinformatics, Centre for Addiction and Mental Health (CAMH), University of Toronto, Toronto, Canada; [5]Rutgers Robert Wood Johnson Medical School, New Brunswick, United States; [6]Institute for Biomedical Engineering, MRI Technology Group, ETH Zürich & University of Zurich, Zurich, Switzerland; [7]Interacting Minds Centre, Aarhus University, Aarhus, Denmark; [8]Scuola Internazionale Superiore di Studi Avanzati (SISSA), Trieste, Italy

**\*For correspondence:**
andreea.diaconescu@utoronto.ca

[†]These authors contributed equally to this work

**Competing interests:** The authors declare that no competing interests exist.

**Abstract** Decision making requires integrating knowledge gathered from personal experiences with advice from others. The neural underpinnings of the process of arbitrating between information sources has not been fully elucidated. In this study, we formalized arbitration as the relative precision of predictions, afforded by each learning system, using hierarchical Bayesian modeling. In a probabilistic learning task, participants predicted the outcome of a lottery using recommendations from a more informed advisor and/or self-sampled outcomes. Decision confidence, as measured by the number of points participants wagered on their predictions, varied with our definition of arbitration as a ratio of precisions. Functional neuroimaging demonstrated that arbitration signals were independent of decision confidence and involved modality-specific brain regions. Arbitrating in favor of self-gathered information activated the dorsolateral prefrontal cortex and the midbrain, whereas arbitrating in favor of social information engaged the ventromedial prefrontal cortex and the amygdala. These findings indicate that relative precision captures arbitration between social and individual learning systems at both behavioral and neural levels.

## Introduction

As social primates navigating an uncertain world, humans use multiple information sources to guide their decisions (*Charness et al., 2013*). For example, in investment decisions, investors may either choose to follow a financial expert's advice about a particular stock or base their decision on their own previous experience with that stock. When information from personal experience and social advice conflict, one source must be favored over the other to guide decision making. We conceptualize the process of selecting between information sources as arbitration. Arbitration is particularly important in uncertain situations when different sources of information have different levels of reliability. While stock performance may fluctuate, the advisor could pursue selfish interests. In our example, investors may track stock performance as it fluctuates and also scrutinize a financial expert's recommendation. Such advice may change based on the advisor's current knowledge and underlying personal incentives. Thus, it is challenging to infer the intentions of the advisor because they are concealed or expressed indirectly, requiring inference from observations of

ambiguous behavior. Optimal arbitration should therefore consider the relative uncertainty associated with each source of information.

Arbitrating between different types of reward predictions based on experiential learning acquired by an individual has been associated with the prefrontal cortex. Specifically, the dorsolateral prefrontal cortex (DLPFC) and the frontopolar cortex have been shown to arbitrate between habitual (model-free) and planned (model-based) learning systems (*Lee et al., 2014*). By contrast, comparatively little is known about how humans weigh self-gathered (individual) reward information against observed (social) information. To investigate this question, we considered two hypotheses: First, arbitration involving social information could rely on theory of mind (ToM) processes, that is inference about others' mental states (*Frith and Frith, 2005*; *Schaafsma et al., 2015*) and higher-level social representations (*Frith, 2012*; *Devaine et al., 2014a*). Accordingly, arbitration involving the intentions of others may rely on activity in classical ToM regions, such as the temporoparietal junction (TPJ) and dorsomedial prefrontal cortex (*Carrington and Bailey, 2009*; *Frith and Frith, 2010*; *Baker, 2011*; *Schurz et al., 2014*). Alternatively, arbitrating between individual and social information may involve similar neural networks as those selecting between model-free and model-based learning (*Lee et al., 2014*), and thus engage lateral prefrontal and frontopolar regions.

It is also worth noting that arbitration depends on both experienced and inferred value learning. Similarly to directly experienced reward learning, inferring on others' intentions engages the striatum, potentially signaling the value associated with social feedback during probabilistic reward learning tasks. For example, parts of the striatum including the caudate show stronger activations in response to reciprocated compared to unreciprocated cooperation during iterative trust games (*Delgado et al., 2005*; *King-Casas et al., 2008*; *Fareri et al., 2015*), and represent social prediction errors signaling a change in fidelity (*Delgado et al., 2005*; *Biele et al., 2009*; *Klucharev et al., 2009*; *Campbell-Meiklejohn et al., 2010*; *Braams et al., 2014*; *Diaconescu et al., 2017*).

In addition, with respect to tracking higher level, contextual change about both reward contingencies and intentionality, one may expect the involvement of the anterior cingulate cortex (ACC). In addition to being associated with volatility tracking in a probabilistic reward learning task (*Behrens et al., 2007*), the ACC was shown to represent volatility precision-weighted prediction errors (PEs) during social learning (*Diaconescu et al., 2017*).

An additional intriguing question is which neuromodulatory system supports the arbitration process. Since arbitration is dependent on the uncertainty of predictions afforded by each learning system, several neuromodulatory systems are good candidates. For non-social forms of learning, previous studies have implicated dopaminergic, cholinergic, and noradrenergic systems in signaling uncertainty, defined as the inverse of precision (*Yu and Dayan, 2005*; *Iglesias et al., 2013*; *Payzan-LeNestour et al., 2013*; *Schwartenbeck et al., 2015*; *Marshall et al., 2016*). Here, we examined how arbitration uniquely modulates activity across dopaminergic, cholinergic, and noradrenergic neuromodulatory systems.

To investigate arbitration between individual and social learning systems, we simulated the aforementioned stock investment scenario in the laboratory. Specifically, we examined how people arbitrate between individual reward information and social advice about a probabilistic lottery where contingencies changed over time. Participants learned to predict the outcome of a binary card draw using advice from a more informed advisor and information inferred from individually observed card outcomes (*Figure 1*).

We separately manipulated the degree of uncertainty (or its inverse, precision) associated with each information source by independently varying the rate of change with which each information source predicted the drawn card color (i.e. volatility; *Behrens et al., 2007*). The advisor was motivated to give correct or incorrect advice depending on the phase of the task, resulting variable reliability of social information. Performing well in the task therefore required participants to track the probabilities of the two sources of information and decide which of the two to trust. We assumed that participants weighed the predictions afforded by each information source as a function of their precision. Thus, we expected participants to rely more on the advice when the advisor's intentions were perceived as stable, and on their personal experience when the intentions of the advisor were perceived to be volatile.

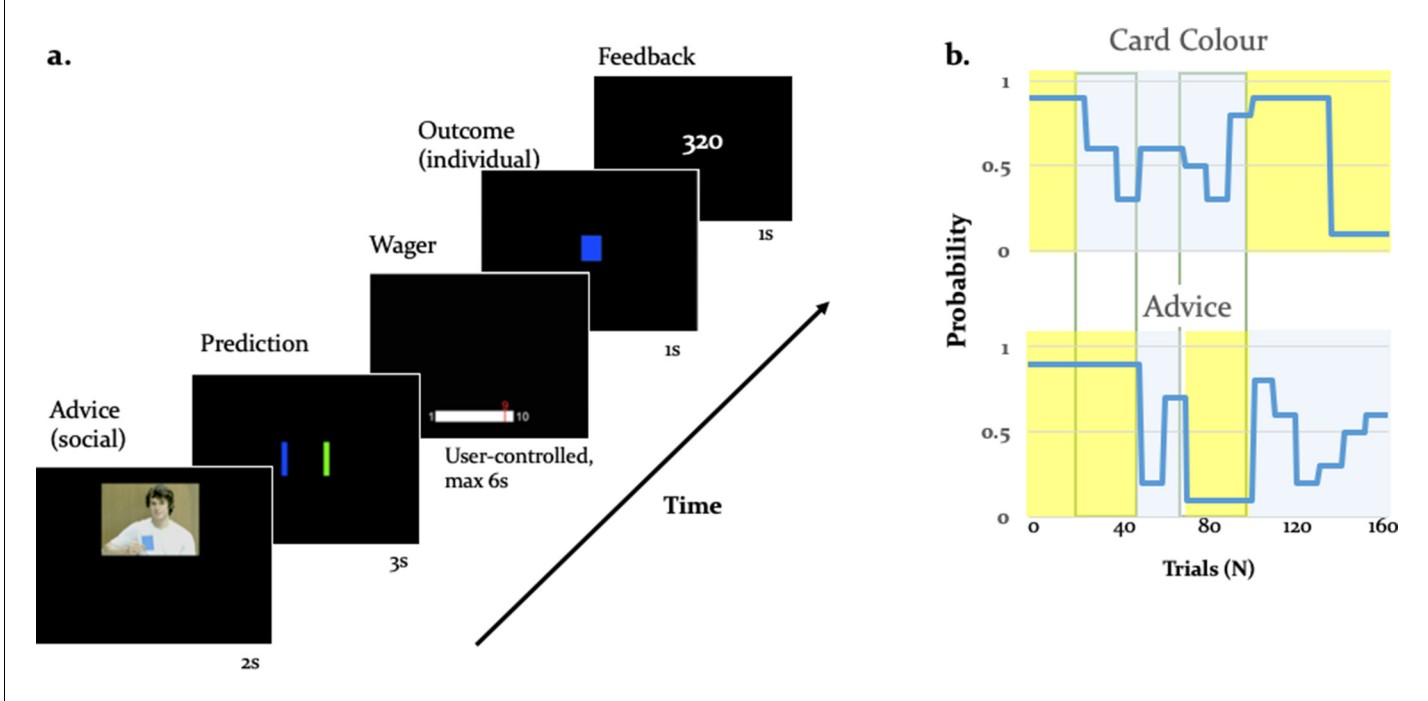

**Figure 1.** Experimental paradigm. (a) Binary lottery game requiring arbitration between individual experience and social information. Volunteers predicted the outcome of a binary lottery, that is whether a blue or green card would be drawn. They could base predictions on two sources of information: advice from a gender-matched advisor (video, presented for 2 s) who was better informed about the color of the drawn card, and on an estimate about the statistical likelihood of the cards being one or the other color that the participant had to infer from own experience (outcome, 1 s). After predicting the color of the rewarded lottery card (user-controlled, maximum 3 s), participants also wagered one to ten points (user-controlled, maximum 6 s), which they would win or lose depending on whether the prediction was right or wrong. After the outcome, participants viewed their cumulative score on the feedback screen (1 s). (b) Contingencies of individual reward and social advice information: Card color probability corresponds to the likelihood of a given color (e.g. blue) being rewarded. The probabilities were matched on average for the two information sources (55% for the card color information and 56% for the advice information). Additionally, the two sources of information were uncorrelated as illustrated by phases of low (yellow) and high (light grey) volatility, enabling a factorial analysis of information source and volatility.

The online version of this article includes the following figure supplement(s) for figure 1:

**Figure supplement 1.** Behavior influenced by volatility.

**Figure supplement 2.** |Average pairwise correlations between regressors.

## Results

To examine the neural mechanisms underlying arbitration, we recruited 48 volunteers (mean age 23.6 ± 1.4, 32 females) to perform a binary lottery task requiring arbitration between individual experienced card outcomes and expert advice. We combined fMRI with a computational modeling approach using the hierarchical Gaussian filter (HGF) (*Mathys et al., 2011*; *Mathys et al., 2014*). This hierarchical Bayesian model is ideally suited to address our question as it examines multilevel inference and provides trial-wise estimates of estimated precision of predictions about each information source. This framework operationalizes arbitration as a precision ratio, corresponding to the relative perceived precision of each information source (*Figure 2*). Thus, arbitration changes as a function of the relative stability of the advice or the card color probabilities. In our paradigm, arbitration increased when the precision of the predictions about one of the two sources of information was high and decreased when both sources were either stable or volatile (see Figure 4 for the arbitration signal averaged across participants).

### Behavior: accuracy of lottery outcome prediction and wager amount

Using the factorial structure of the task, we tested the impact of volatility on performance with a two-factor repeated measures ANOVA, where the two factors were information source (card versus advice) and phase (stable versus volatile). Across all behavioral metrics, we observed an effect of

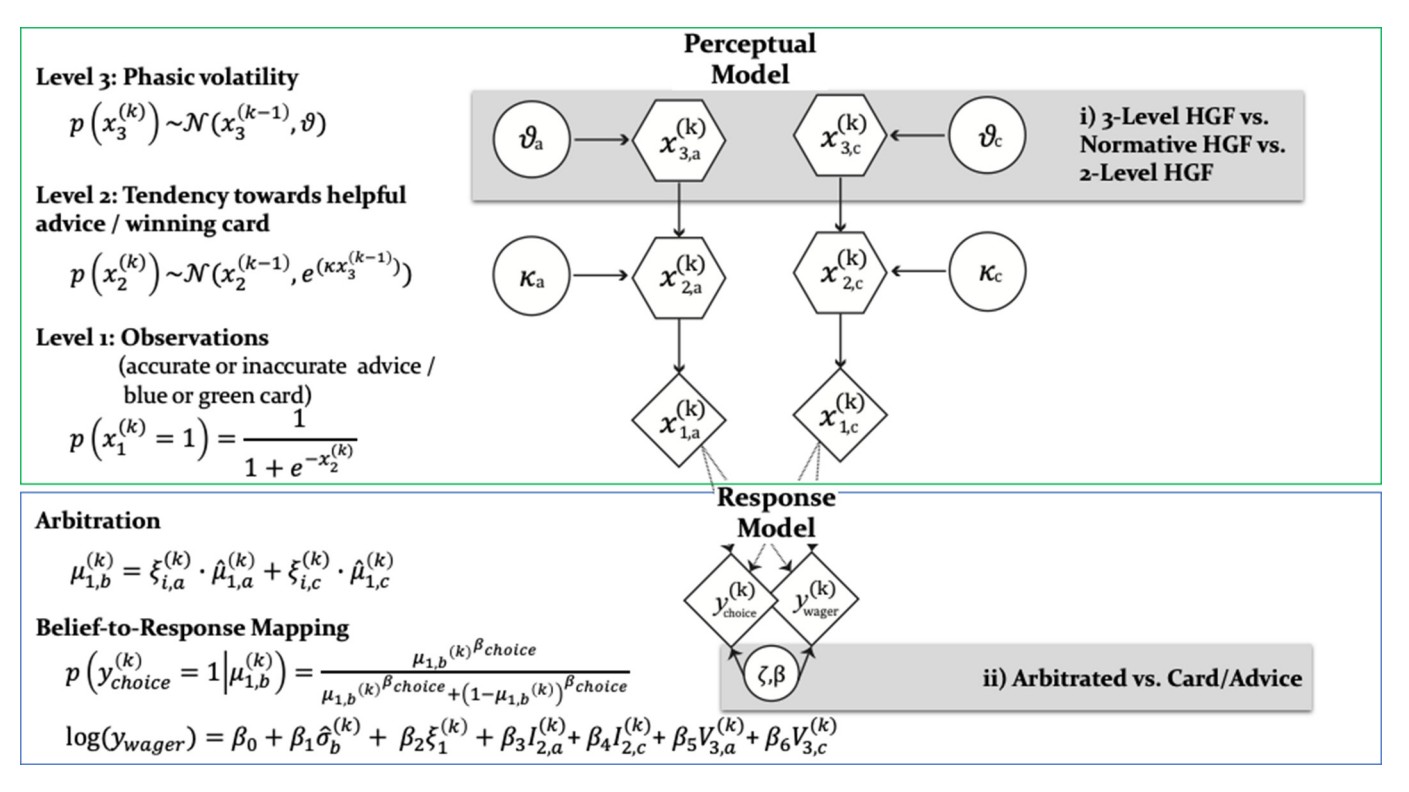

**Figure 2.** Computational learning and arbitration model. In this graphical notation, circles represent constants whereas hexagons and diamonds represent quantities that change in time (i.e. that carry a time/trial index). Hexagons in contrast to diamonds additionally depend on the previous state in time in a Markovian fashion. The two-branch HGF describes the generative model for advice and card probability: $x_1$ represents the accuracy of the current advice/card color probability, $x_2$ the tendency of the advisor to offer helpful advice tendency of card color to be rewarded, and $x_3$ the current volatility of the advisor's intentions/card color probabilities. Learning parameters describe how the states evolve in time. Parameter $\kappa$ determines how strongly $x_2$ and $x_3$ are coupled, and $\vartheta$ represents the meta-volatility of $x_3$. The response model maps the predicted color probabilities to choices. The response model also assumes that trial-wise wagers and predictions arise from a linear combination of arbitration, informational uncertainty (advice and card), and volatility (advice and card). For model selection, we combined three perception with three response models (see *Figure 3*). All the models considered can be grouped according to common features and divided into model families: (i) the Perceptual model families distinguish between more (non-normative and normative three-level) and less (two-level) complex types of HGFs. More specifically, the distinction between three-level and two-level HGFs refers to estimating or fixing the volatility of the third level; normative in contrast to non-normative HGFs assume optimal Bayesian inference. (ii) Response model families distinguish between arbitrated and single-information source – advice or card only – models, which correspond to estimating parameter $\vartheta$ or fixing it to reduce arbitration to either the advice prediction or the card color prediction.

The online version of this article includes the following figure supplement(s) for figure 2:

**Figure supplement 1.** Parameter recovery when using empirical parameter values (Binary HGF).

phase, indicating a reduction in performance in volatile compared to stable phases, and a phase × information interaction, indicating that the effect was larger for the social than the individual source of information. First, for the accuracy with which participants predicted lottery outcome, we found a main effect of phase ($df = (1,36)$, $F = 187.94$, $p = 7.7e\text{-}16$) and an information source-by-phase interaction ($df = (1,36)$, $F = 11.13$, $p = 0.0020$) (see *Figure 1—figure supplement 1a*). Thus, in-keeping with the rationale that arbitration relates to relative information quality, the degree to which participants relied on each information source was a function of precision as manipulated using the volatility structure of the task. Participants performed significantly better in stable compared to volatile periods of the task. These effects were not modulated by fatigue, as we found no significant differences between early and late phases of the task.

Second, advice-taking behavior differed as a function of volatility and information source: For the percentage of trials in which participants followed a given source of information, we detected a main effect of phase ($df = (1,36)$, $F = 56.26$, $p=7.3073e\text{-}09$) and an information source-by-phase

interaction ($df$ = (1,36), $F$ = 25.86, p=1.1561e-05) (*Figure 1—figure supplement 1b*). Thus, participants took advice less often particularly when it was volatile rather than stable.

Third, the amount of points wagered also depended on the task volatility and the information source. We observed a main effect of phase ($df$ = (1,36), $F$ = 28.78, $p$ = 4.54e-06) and an information source-by-phase interaction ($df$ = (1,36), $F$ = 16.75, $p$ = 2.21e-04; *Figure 1—figure supplement 1c*). Participants wagered fewer points particularly when advice was volatile. Moreover, the number of points wagered correlated significantly with the total score in stable phases ($r$ = 0.37, $p$ = 0.02), but not in volatile phases ($r$ = 0.30, $p$ = 0.06). Simulations using a two-level HGF (with low and fixed volatility) suggested that tracking volatility is beneficial for task performance: a hypothetical person who did not take the volatility of the task phases into account gained on average 21.6 points less than an agent tracking volatility. In line with previous evidence (*Behrens et al., 2008*), these results emphasize the impact of volatility on the willingness to invest and investment success as measured here by total score.

## Advisor ratings

Participants were asked to rate the advisor (i.e. helpful, misleading, or neutral with regard to suggesting the correct outcome) in a multiple-choice question presented five times during the experiment. The time points were associated with different social and individual information (initial/prior: 1st trial; stable advice, stable card phase = (14th trial); stable advice, volatile card phase (49th trial); volatile advice, volatile card phase (73rd trial); volatile advice, stable card phase = 115th trial). On average, participants rated the advice as 75.0 ± 4.6% (mean ± standard deviation) helpful in the stable advice phase. The corresponding values were 50 ± 3.4% in the volatile advice phase, 63.8 ± 4.4% in the stable card phase, and 61.2 ± 3.8% in the volatile card phase.

We examined the extent to which participants' ratings changed as a function of the task phases, and found a significant main effect of phase ($df$ = (1,36), $F$ = 15.67, $p$ = 3.3e-04) and a significant information source × phase interaction ($df$ = (1,36), $F$ = 8.42, $p$ = 0.0062). This suggests that advice ratings decreased during volatile compared to stable phases, and this effect was more strongly related to the advice compared to the card information.

## Debriefing questionnaire

After completing the task, participants filled out a task-specific debriefing questionnaire, assessing their perception of the advisor and how they integrated the social information during the task. The questions were originally presented to participants in their native German, and are translated here into English.

First, participants were asked to describe the strategy the advisor used in the game (debriefing question 3: *'Did the advisor intentionally use a strategy during the task? If yes, describe what strategy that was')*. Thirty out of 38 participants answered 'Yes' to this question, and described (in their own words) the advisor's strategy. We repeated our analyses including only these 30 participants and found that all conclusions remained statistically the same. Second, participants were asked to rate the advice on a 6-point Likert scale ranging from unhelpful to very helpful (debriefing question 4: *'How helpful did you perceive the advice you received?"*). In general, participants rated the advisors' recommendations as helpful (mean ratings 4.2 ± 1.0, ranging from 2 to 6). Finally, we also asked participants to rate, in terms of percentages, how often they followed the advice (debriefing question 5: *'How often did you follow the recommendations of the advisor?"*). On average, participants reported that they followed the advice 60% of the time (mean ratings 60 ± 12), which significantly differed from chance (t(37) = 5.02, p=1.29e-05). Thus, participants experienced advisors as intentional and helpful, which are core characteristics of social agents.

## Model-based results

We used computational modeling with hierarchical Gaussian Filters (HGF; *Figure 2*) to explain participants' responses on every trial. To contrast competing mechanisms underlying learning and arbitration, our model space included a total of nine models (*Figure 3a*). Non-normative perceptual models varied in complexity of volatility processing (three-level full HGF vs. two-level no-volatility HGF), normative perceptual models assumed optimal Bayesian inference (normative HGF), and response models varied in the extent of arbitration (arbitration; no arbitration: advice only; no

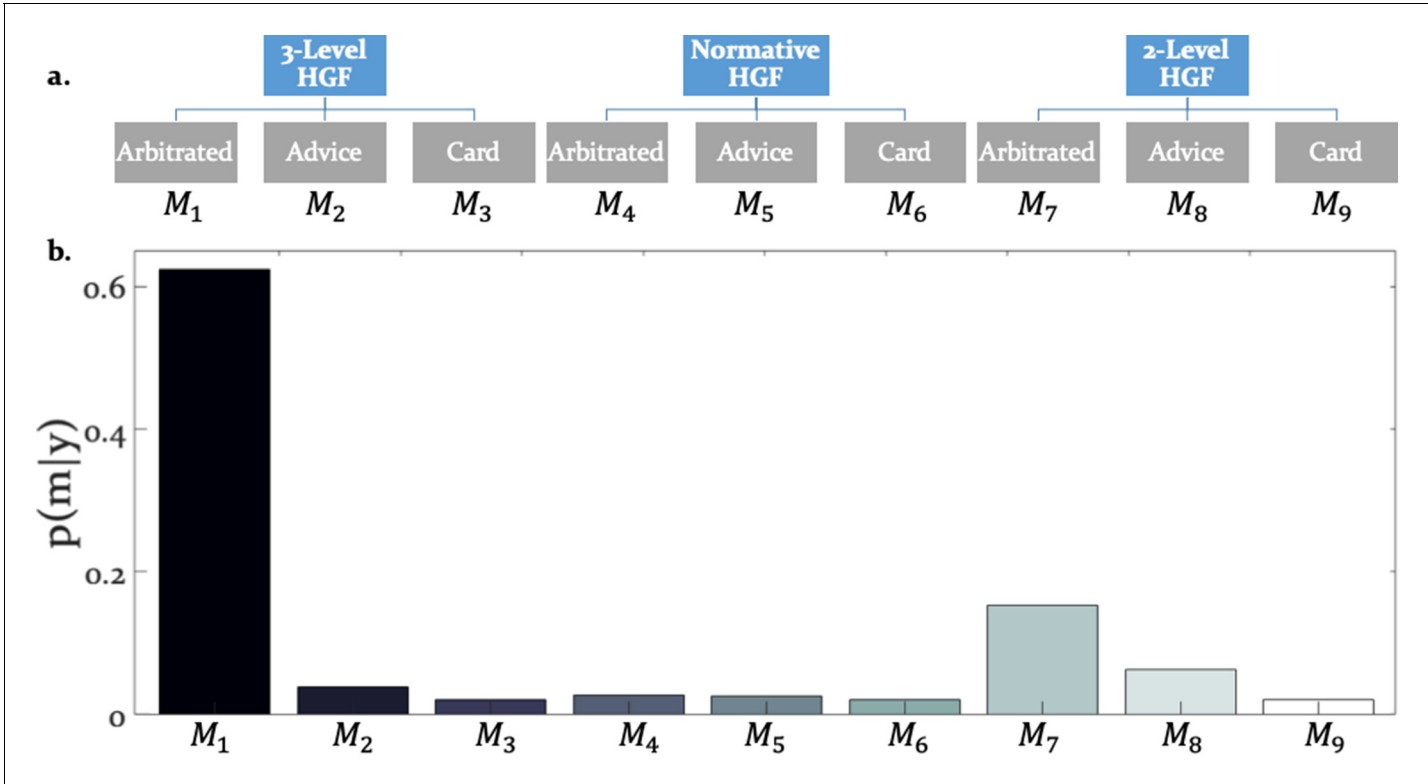

**Figure 3.** Hierarchical structure of the model space and model selection results. (**a**) The learning and arbitration models considered in this study have a 3 × 3 factorial structure and can be displayed as a tree. The nodes at the top level represent the perceptual model families (three-level HGF, normative HGF, two-level non-volatility HGF). The leaves at the bottom represent response models which integrate and arbitrate between social and individual sources of information ('Arbitrated') or exclusively consider social ('Advice') or individual ('Card') information. (**b**) Random effects Bayesian model selection revealed one winning model, the Arbitrated three-level HGF. Posterior model probabilities or $p(m|y)$ indicated that this model best explained participants' behavior in the majority of the cases.

arbitration: card information only). Bayesian model selection (*Stephan et al., 2009*) served to compare models (see Materials and methods and *Figure 2* for details). For model comparison, we used the log model evidence (LME), which represents a trade-off between model complexity and model fit.

## Do participants arbitrate between advice and individually sampled card outcomes?

The winning model was the three-level HGF with arbitration ($\phi_p$ = 0.999; Bayes Omnibus Risk = 4.26e-11; *Figure 3b*; *Table 1a*). This model formalised arbitration as a ratio of precisions: the precision of the prediction about advice accuracy and color probability, divided by total precision. Moreover, the model included a social bias parameter reflecting the degree to which participants followed the advisor irrespective of task information. The model family that included volatility of both information sources outperformed models without volatility, in-keeping with the model-independent finding that perceived volatility of both information sources affected behavior.

## Is the parameter estimation robust?

The winning three-level full HGF model includes multiple parameters that need to be estimated. A general question is whether these parameters are 'practically identifiable', that is whether their values can be recovered accurately given the actual experimental design. To examine this question, we simulated responses based on all participants' maximum-a-posteriori estimates of the parameters, and then fitted the model to those simulated responses in order to test whether we could recover the same parameter estimates.

**Table 1.** (a) Results of Bayesian model selection: Model probability ($p(m|y)$) and protected exceedance probabilities ($\phi_p$).

Please refer to the participants' LME and BMS results in **Table 1—source datas 1** and **2**, respectively. (b) Average maximum a-posteriori estimates of the learning and arbitration parameters of the winning model (Arbitrated three-level HGF). Please refer to participants' individual posterior parameter estimates for perceptual and response model parameters in **Table 1—source datas 3** and **4**.

| | Perceptual Models: | | |
|---|---|---|---|
| Response models: | Arbitrated | Advice Only | Card Only |
| | | 3-level HGF | |
| $p(m|y)$ | 0.63 | 0.04 | 0.02 |
| $\phi_p$ | 0.99 | 4.7e-12 | 4.7e-12 |
| | | Normative HGF | |
| $p(m|y)$ | 0.03 | 0.03 | 0.02 |
| $\phi_p$ | 4.7e-12 | 4.7e-12 | 4.7e-12 |
| | | 2-level HGF | |
| $p(m|y)$ | 0.15 | 0.06 | 0.02 |
| $\phi_p$ | 6.2e-05 | 4.7e-12 | 4.7e-12 |

| Perceptual Model Parameters | Mean | SD | Response Model Parameters | Mean | SD |
|---|---|---|---|---|---|
| $\kappa_c$ | 0.58 | 0.17 | $\zeta$ | 1.03 | 1.24 |
| $\kappa_a$ | 0.56 | 0.28 | $\beta_1$ | −1.59 | 0.94 |
| $\vartheta_c$ | 0.59 | 0.07 | $\beta_2$ | 1.42 | 1.69 |
| $\vartheta_a$ | 0.62 | 0.09 | $\beta_3$ | 0.23 | 1.37 |
| | | | $\beta_4$ | 0.63 | 1.24 |
| | | | $\beta_5$ | −2.97 | 2.47 |
| | | | $\beta_6$ | −0.51 | 1.83 |
| | | | $\beta_{ch}$ | 2.25 | 0.92 |

The online version of this article includes the following source data for Table 1:

**Source data 1.** Log model evidences for all models.

**Source data 2.** Random effects Bayesian model selection.

**Source data 3.** Maximum a posteriori estimates of the perceptual model parameters and response model parameters influencing choice along with subject IDs.

**Source data 4.** Maximum a posteriori estimates of the response model parameters influencing wagers along with subject IDs.

To assess and compare degrees of parameter recovery, we categorized it in terms of effect sizes, that is, whether the relationship between the original and the recovered values indicates small, medium, or large effect sizes as quantified by Cohen's $f$. For a multiple regression analysis, a Cohen's $f$ above 0.4 is conventionally regarded as a large effect size. Based on this criterion, we could recover all parameters well, as all Cohen's $f$ values equaled or exceeded 0.4 (see **Figure 2— figure supplement 1**).

## Do participants differ in how they learn from advice and use it to predict lottery outcomes?

Three parameters modulated the arbitration signal of the winning model. These included: (i) $\kappa$ or the coupling between the two hierarchical levels that determined the impact of volatility on the inferred predictions of each information source (**Equation 6**), (ii) $\vartheta$, determining the variance of the volatility (**Equation 12**), and (iii) $\zeta$, the social bias which reflected the reliance on the advice independent of its reliability (**Equation 19**). Both coupling $\kappa$ and volatility parameter $\vartheta$ did not differ significantly

between learning from individual and social information (t(36) = 0.28, p=0.77 for $\kappa$ and t(36) = -1.59, p=0.12 for $\vartheta$; *Figure 4a-b*). In fact, they were highly correlated: $r_1$=0.55, $p_1$=0.003 for $\kappa$ and $r_2$=0.64, $p_2$=0.001 for $\vartheta$. This result suggests that participants learned similarly from individual (volatile card probabilities) and social (advisor fidelity) information.

The reliability-independent social bias parameter $\zeta$ differed significantly from zero (t(36) = 5.09, p=1.07e-05). Importantly, since the social bias parameter $\zeta$ is coded in log-space, the prior value of zero refers to a uniform weighting of the two cues in linear parameter space. Thus, on average, participants relied more on the advisor's recommendations compared to their own sampling of the card outcomes (*Figure 4c*).

## Do the response model parameter estimates explain wager behavior?

Decisions of how many points participants were willing to wager on a given trial (a measure of confidence) were related to several model-based quantities, including (irreducible) uncertainty of the agent's beliefs about the decision, arbitration, and the estimated volatility of the advisor's intentions (belief uncertainty: t(37) = -10.37, $p_{bonf}$ = 1.0e-11; arbitration: t(37)=5.16, $p_{bonf}$ = 5e-05; and estimated advisor volatility: t(37)=-7.41 $p_{bonf}$ = 4.75e-08) (*Figure 5*). The stronger the bias to arbitrate in favor of social information, the more points participants wagered. Conversely, estimated advisor volatility was negatively associated with the amount wagered: the higher the estimated advisor volatility, the fewer points participants were willing to wager on a given trial (see *Table 2* for the priors over the parameters, *Table 1b* for all parameter estimates, and *Figure 5* for the trial-wise influence of the average computational quantities on wager amount).

## Do the model parameter estimates explain perceived advice accuracy and wager amount?

We aimed to examine at the behavioral level whether the model predictions were consistent with participants' perceptions of the advice accuracy during the experiment. Participants judged advice accuracy (i.e. helpful, misleading, or neutral with regard to predicting actual card color) in a multiple-choice question presented 5fivetimes during the experiment (initial/prior: 1st trial; stable advice, stable card phase = (14th trial); stable advice, volatile card phase (49th trial); volatile advice, volatile card phase (73rd trial); volatile advice, stable card phase = 115th trial). We first tested whether the responses to these questions positively related to estimates of advice accuracy $(\mu_{1,a}^{(k)})$ that were extracted from the winning model. A linear regression analysis demonstrated that the inferred advice accuracy or $\mu_{1,a}^{(k)}$ measured at the time of the multiple-choice question, predicted participants' selections. Specifically, the estimated beta parameter estimate across all task phases was significantly different from zero (t(36) = 4.71, p=3e-05). These findings suggest that the model predicted independently (and discretely) measured perception of advice accuracy, in-keeping with the internal validity of the model.

Next, we tested whether the wager amounts predicted by the model correlated with participants' actual wagers. In all four conditions of the task, the predicted wager significantly correlated with the number of points participants actually wagered: (i) advice stable phase $r_1$ = 0.62, $p_1$ = 3e-05; (ii) advice volatile phase $r_2$ = 0.63, $p_2$ = 2e-05; (iii) card stable phase $r_3$ = 0.81, $p_4$ = 9e-10; and (iv) card volatile phase $r_4$ = 0.80, $p_4$ = 1e-09 (*Figure 5—figure supplement 1*). These findings suggest that the winning model explained variation in (the continuously measured) actual wager amount.

## Do the model parameter estimates explain participants' self-reports?

We used classical multiple regression and post-hoc tests to examine whether the model parameter estimates extracted from the winning model ($M_1$) explained participants' advisor ratings, as measured by debriefing questions after the main experiment outside the scanner. Participants who reported that the advisor intentionally tried to help or mislead at different phases of the task showed a trend towards a larger estimate of the social weighting parameter $\zeta$ (df = (1,36), F = 3.49, p = 0.06). Moreover, advice helpfulness ratings were explained by model parameter estimates ($R^2$ = 32.2%, F = 2.46, p=0.04). This effect was primarily driven by parameter $\kappa_a$ (r(37)=0.47, p=0.0026), indicating that participants who rated the advice as being helpful showed stronger coupling between two levels of the hierarchical model. More specifically, participants who rated the advice as more helpful displayed higher $\kappa_a$ values, that is, increased sensitivity to the changing phases of

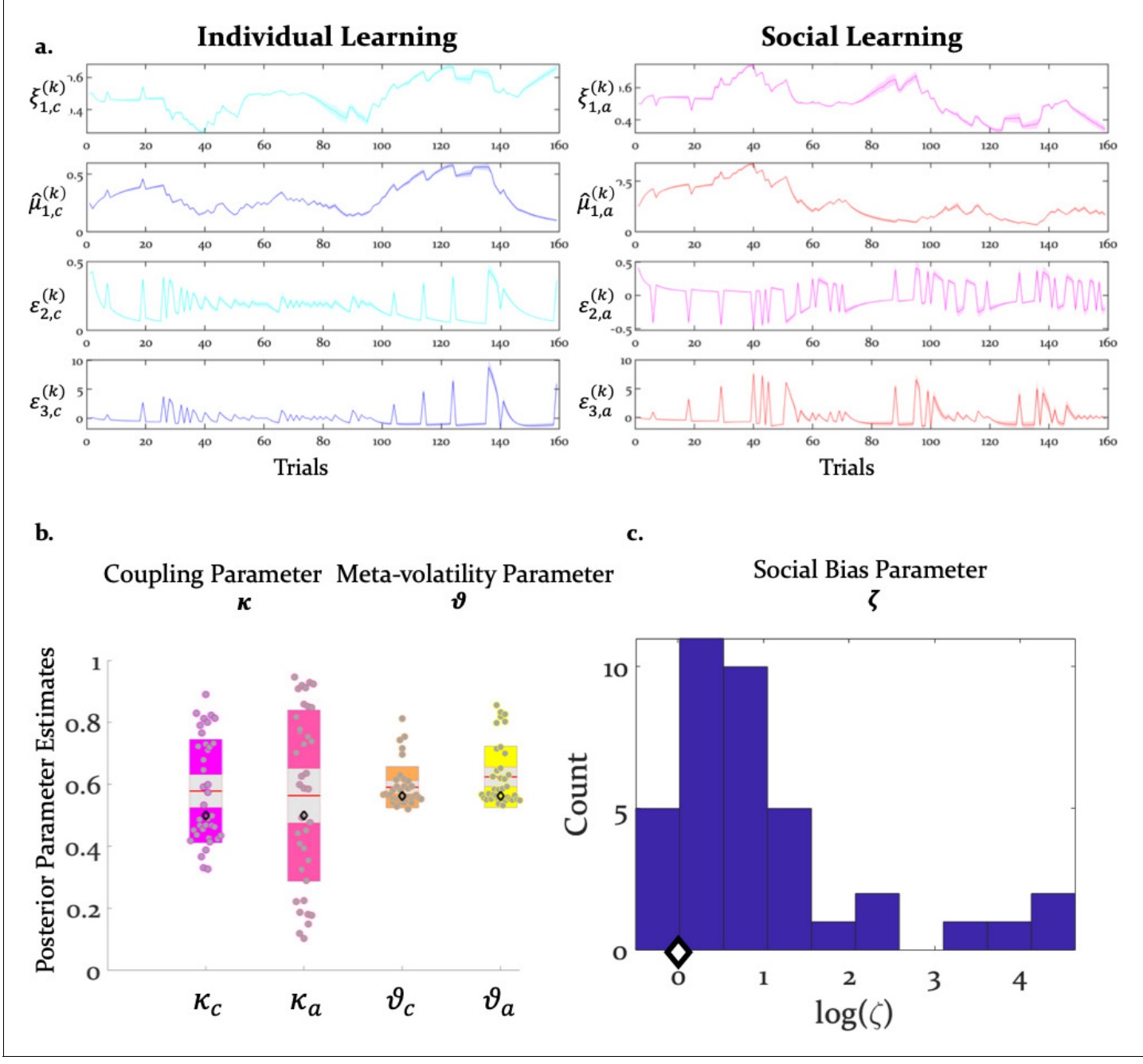

**Figure 4.** Inference and arbitration of individual and social learning. (a) Average trajectories for arbitration and hierarchical precision-weighted PEs for individual and social learning (see Materials and methods for the exact equations): $\xi_a$ = arbitration in favor of the advice (*Equation 19*); $\xi_c$ = arbitration in favor of individually estimated card color probability (*Equation 20*). $\mu_{1,a}$ = estimated advice accuracy (*Equation 4*); $\mu_{1,c}$ = individually estimated card color probability (*Equation 18*). $\varepsilon_{2,a}$ = precision-weighted prediction error (PE) of advisor fidelity (*Equation 8*); $\varepsilon_{2,c}$ = unsigned (absolute) precision-weighted PE of card outcome (absolute value of *Equation 14*). $\varepsilon_{3,a}$ = precision-weighted advice volatility PE (*Equation 13*); $\varepsilon_{3,c}$ = precision-weighted card color volatility PE (*Equation 15*). Line plots were generated by averaging the computational trajectories of the winning (Arbitrated 3-HGF: *Figure 2*) model across all participants for each of the 160 trials. The shaded area around each line depicts +/- standard error of the mean over participants. (b) Group means, standard deviations and prior values for the perceptual model parameters determining dynamics of computational trajectories in (a). Jittered participant-specific estimates are plotted for each perceptual model parameter, red lines indicate the group mean, grey areas reflect 1 SD of the mean, and colored areas the 95% confidence intervals of the mean. (c) Distribution of $\log(\zeta)$ values. In (b) and (c), black diamonds denote the priors of each parameter (for details, see *Table 2*).

**Table 2.** Prior mean and variance of the perceptual and response model parameters.

| Model | | Prior mean | Prior variance |
|---|---|---|---|
| Perceptual models: | | | |
| Three-level HGF | $\kappa_a, \kappa_c$ | 0.5 | 1 |
| | $\vartheta_a, \vartheta_c$ | 0.55 | 1 |
| Normative HGF | $\kappa_a, \kappa_c$ | 0.5 | 0 |
| | $\vartheta_a, \vartheta_c$ | 0.55 | 0 |
| Two-level HGF | $\vartheta_a, \vartheta_c$ | 0.00062 | 0 |
| | | | |
| Response models: | | | |
| | $\beta_{1-6}$ | 0 | 4 |
| | $\beta_{ch}$ | 48 | 1 |
| | $\beta_0$ | 6.21 | 4 |
| | $\beta_{wager}$ | 1.50 | 100 |
| 1. Arbitrated | $\zeta$ | 0 | 25 |
| 2. Advice Only | $\zeta$ | Inf | 0 |
| 3. Card Only | $\zeta$ | 0 | 0 |

Note: The prior variances are given in the numeric space in which parameters are estimated. $\kappa$, $\vartheta$, and $\mu_3^{(k=0)}$ are estimated in logit-space, while the other parameters are estimated in log-space. Although the prior variances for all parameters are set to be rather broad, we selected a shrinkage prior mean and variance for the decision noise parameter $\beta_{ch}$ such that behavior is explained more by variations in the remaining parameters rather than decision noise.

advice validity, adjusting their wagering behavior more strongly to the advisor's strategy. Thus, not only did the participants perceive the advice in our task as intentional and helpful, our model also explained some of these impressions.

## Neural signatures of arbitration

Using behaviorally fitted computational trajectories to generate participant-specific GLMs for model-based fMRI analysis, we examined how the brain arbitrates between social and individual learning systems. We conceptualized the learning and arbitration process as hierarchical Bayesian inference, and fitted the participant-specific trajectories that reflect arbitration (*Equation 20*) to fMRI data.

Hierarchical precision-weighted PE signals were replicated in the same dopaminergic and fronto-parietal regions as in previous studies using other sensory and social learning domains (see *Iglesias et al., 2013*; *Diaconescu et al., 2017*), indicating that the modifications in the experimental paradigm did not affect basic learning processes (see *Figure 6—figure supplements 1–2*).

Undirected tests for arbitration activity identified ventral prefrontal regions, such as the left ventromedial PFC (peak at [-2, 46,–10]) and the right orbitofrontal cortex (OFC) [26, 34, -10]. Interestingly, frontal activations also included the right frontopolar cortex [4, 54, 30] and ventrolateral prefrontal cortex (VLPFC) [50, 36, 0], regions previously associated with arbitration between model-based and model-free forms of individual learning (*Lee et al., 2014*; *Figure 6*). The right VLPFC showing arbitration-related effects at [48, 35, -2] significantly overlapped with the arbitration-related reliability activations detected by Lee and colleagues, supporting the notion that arbitration is to some extent domain-independent.

In addition, we found that a wide network of cortical and subcortical regions contributes to arbitration that included occipital areas, the anterior insula, left thalamus, left putamen, bilateral middle cingulate sulcus, supplementary motor area (SMA) [−2, -8, 52], left dorsal middle cingulate gyrus [−10,–26, 44], the right amygdala [18, -10, -16] and the left midbrain [−6,–18, −12] (*Table 3*, *Figure 6*). Thus, a network of cortical and subcortical regions contributed to arbitration.

Directed tests for arbitration in favor of individual over social information identified activity increases in the right dorsolateral PFC [36, 46, 30], left SMA/anterior cingulate sulcus [−2,–8, 52] and

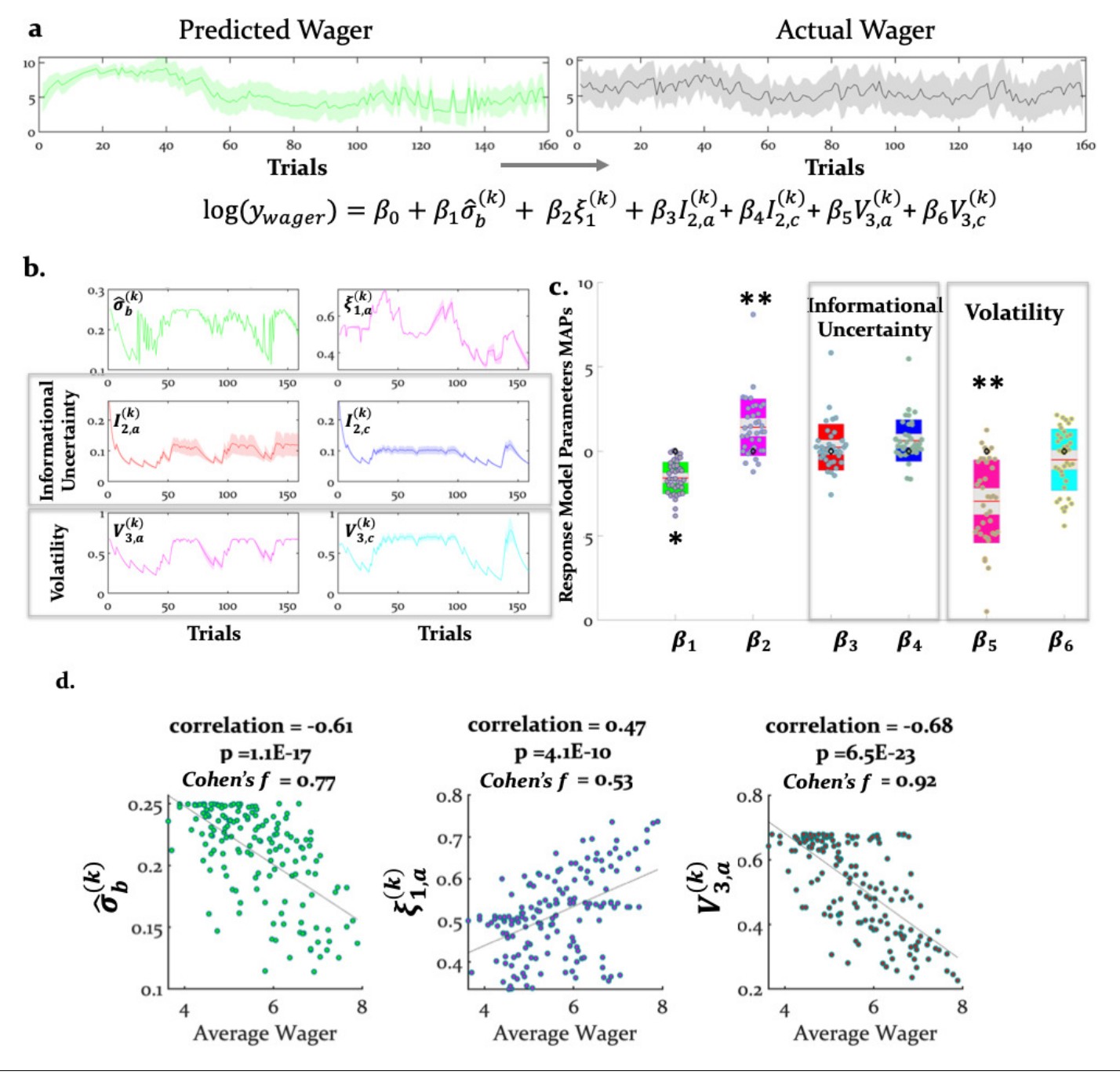

**Figure 5.** Computational quantities and model parameters explaining wager amount. (a) With our response model, we predicted that the actual trial-wise wager (right) could be explained (left and bottom) by the six key trajectories (see *Equation 21*) given in (b). These include (i) (irreducible) belief uncertainty (based on the integrated belief of individual and advice predictions; *Equation 24*); (ii) arbitration in favour of advice (*Equation 19*); (iii) informational uncertainty (*Equation 25*) and volatility of the advice (*Equation 26*) and (iv) informational uncertainty and volatility of the card (same *Equations 25 and 26*, but for the card modality). (a) and (b) show group averages (see Materials and methods for the exact equations). For the model-based parameters, the line plots were generated by averaging the computational trajectories of the winning (Arbitrated 3-HGF: *Figure 2*) model across all participants for each of the 160 trials. The shaded areas depict +/- standard error of this mean over participants. (c) Group means, standard deviations and prior values for the response model parameters determining the impact of those trajectories (i.e. uncertainties and arbitration) on trial-wise wager amount. Jittered raw data are plotted for each parameter. Red lines indicate the mean, grey areas reflect 1 SD from the mean, and the colored areas the 95% confidence intervals of the mean. The black diamonds denote the prior of the parameters, which in this case is zero. $^*p<0.05$, $^{**}p<0.001$. (d) Scatter plots with average actual wager on the x-axis and average of the computational variables assumed to impact the trial-wise wager: belief uncertainty, arbitration in favor of advice, and volatility of advice on the y-axes, respectively. The correlation coefficients (with corresponding p

*Figure 5 continued on next page*

*Figure 5 continued*

values), regression slopes, and effect sizes (Cohen's *f*) are included to quantify the relationship between the actual wager and the computational quantities that showed a significant relation to wagers.

The online version of this article includes the following figure supplement(s) for figure 5:

**Figure supplement 1.** Model validity with regard to wager amount.

the midbrain [−6,–18,−12] (*Figure 7a*). The BOLD signal change in these regions peaked during the time window of the wager decision. In summary, primarily dorsal regions of PFC were modulated by arbitration in favor of individually estimated card probability.

Conversely, activity in the right amygdala, VLPFC, orbitofrontal and ventromedial PFC was modulated by arbitration in favor of the advisor's suggestions (*Figure 7b*). Outside PFC, the right anterior TPJ [56, -52, 24], right superior temporal gyrus [52, -18, -8], and right precuneus [6, -51, 32] showed similar effects (*Tables 4* and *5* for the entire list of brain regions). Thus, primarily ventral regions of PFC together with temporal and parietal regions were more active during arbitration in favor of social information.

To examine effects of arbitration in dopaminergic, cholinergic, and noradrenergic regions, we also performed region-of-interest (ROI) analyses using a combined anatomical mask of dopaminergic, cholinergic, and noradrenergic nuclei. A single cluster in the right substantia nigra survived small-volume correction ($p<0.05$ FWE voxel-level corrected for the entire ROI; peak at [−6,–18,

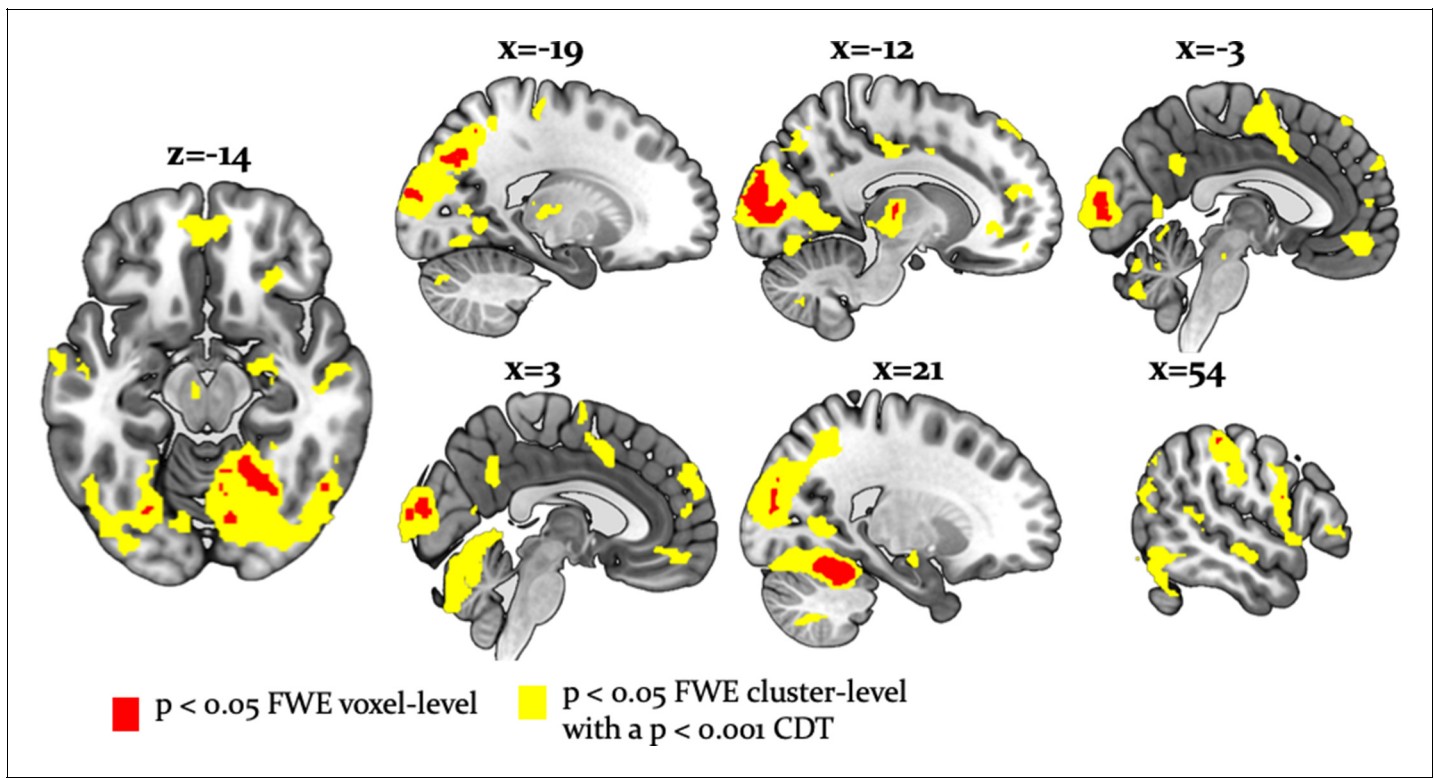

**Figure 6.** Whole-brain undirected arbitration signals. Effects of arbitration in favor of one or the other source of information were detected in ventromedial PFC, orbitofrontal cortex, right frontopolar cortex, VLPFC, the left midbrain, bilateral fusiform gyrus, lateral occipital gyrus, lingual gyrus, anterior insula, right amygdala, left thalamus, right cerebellum, bilateral middle cingulate sulcus and SMA. The figure shows whole-brain FWE-corrected voxel (red) - and cluster-level-corrected (yellow) results of an undirected *F*-test, $p<0.05$ (CDT = cluster defining voxel-level threshold).

The online version of this article includes the following figure supplement(s) for figure 6:

**Figure supplement 1.** Main effects of precision-weighted PEs about card and advice outcomes (*Equations 8 and 14*).

**Figure supplement 2.** Main effects of precision-weighted PEs about card and advice volatility.

**Table 3.** MNI coordinates and F-statistic of maxima of activations induced by either form of arbitration (**Equations 19-20**; p<0.05, cluster-level whole-brain FWE corrected).
Related to **Figure 7**.

| | Hemisphere | X | Y | Z | # Voxels | *F-statistic* |
|---|---|---|---|---|---|---|
| $\xi^{(k)}$ | | | | | | |
| Midbrain | L | -6 | −18 | −12 | 20 | 23.49 |
| Thalamus | L | −12 | −18 | 8 | 490 | 59.87 |
| Anterior insula | L | −44 | 2 | 0 | 1744 | 52.97 |
| Anterior insula | R | 48 | 6 | -2 | 813 | 31.56 |
| Fusiform gyrus | R | 28 | −78 | −10 | 1327 | 75.32 |
| Fusiform gyrus | L | −28 | −76 | −10 | 227 | 39.55 |
| Inferior occipital gyrus | R | 48 | −68 | −10 | 810 | 52.70 |
| Inferior occipital gyrus | L | −42 | −68 | -4 | 1519 | 67.56 |
| Calcarine sulcus | R | 12 | −86 | 6 | 22285 | 199.99 |
| Superior temporal gyrus | L | −60 | −30 | -2 | 79 | 24.02 |
| Superior temporal sulcus | R | 52 | −18 | -8 | 104 | 30.35 |
| Amygdala | R | 18 | −10 | −16 | 76 | 27.01 |
| Precuneus | R | 4 | −52 | 30 | 238 | 38.50 |
| Dorsal medial PFC | L | −10 | 44 | 52 | 108 | 23.14 |
| Superior medial PFC | R | 4 | 56 | 28 | 493 | 39.83 |
| Ventrolateral PFC | R | 50 | 36 | 0 | 202 | 24.28 |
| Frontopolar cortex | R | 4 | 54 | 30 | 138 | 24.28 |
| Orbitofrontal cortex | R | 26 | 34 | −10 | 80 | 30.47 |
| Ventromedial PFC | L | -2 | 46 | −10 | 393 | 37.43 |
| Supramarginal gyrus | R | 54 | −30 | 50 | 46.46 | 952 |
| Cerebellum | R | 18 | −48 | −18 | 1919 | 166.69 |

−12]; *Figure 8*). Activity in this region increased with arbitration in favor of individual estimates of card probabilities rather than advice.

It is important to note that these regions showed significantly larger effects of arbitration than of the amount of points wagered. Responses reflecting arbitration dominated over responses reflecting wager amount in cerebellar, midbrain, occipital, parietal, medial prefrontal, and temporal regions including the amygdala. Activity in precuneus and ventromedial prefrontal cortex in turn correlated with wager amount (*Figure 9*). As wager amount can be taken as a proxy for decision value or confidence (*Lebreton et al., 2015*), these data suggest that arbitration signals arise on top of decision value and confidence. Moreover, we captured arbitration as a model-derived, continuous, and time-resolved variable. Thus, our findings elucidate the process rather than the result of arbitration.

## Main effect of stability and interaction with source of information

To examine arbitration from a different angle, we also conducted a factorial analysis. This was possible because we employed a 2 × 2 factorial design – that is, two sources of information (individual versus social) in two different states (stable versus volatile) (*Figure 10a*). Specifically, we contrasted volatile with stable phases across both information modalities. Volatility is closely tied to arbitration because it potentiates the perceived uncertainty associated with a given information source, and thereby the need to arbitrate. We assumed that arbitration increased when one of the two information sources was perceived as being more stable than the other. In all comparisons, we controlled for decision value and confidence by using the trial-wise wager amount as a parametric modulator in the analysis of brain data. We found two significant results (*Figure 10b*): (i) a main effect of task

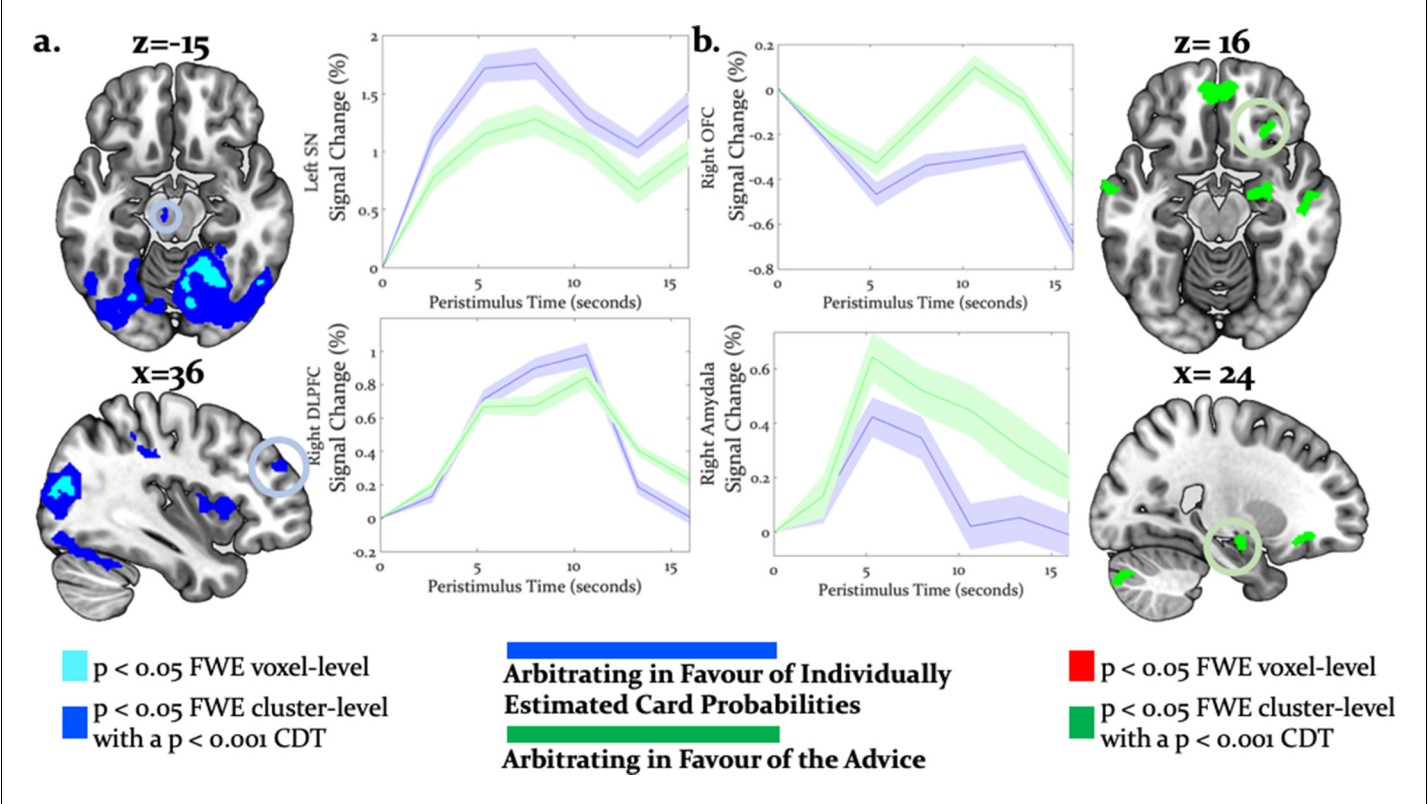

**Figure 7.** Neural arbitration directed to specific source of information. (a) Activity in the left midbrain (substantia nigra (SN)) [−6,–18, −10] (top) and the right DLPFC [36, 46, 30] (bottom) during the prediction of card color increased more when participants arbitrated in favor of individually estimated card color probability as compared to the advisor's suggestions (whole-brain FWE cluster-level corrected, p<0.05). (b) Activity in right (OFC [28, 26, -16] (top) and in right amygdala [18, -10, -16] (bottom) increased more when participants arbitrated in favor of the advisor's suggestion than when they arbitrated in favor of the individually learned estimates of card probability (whole-brain FWE cluster-level corrected, p<0.05). The line plots reflect the average BOLD signal activity in the respective significantly activated cluster aligned to the onset of advice presentation relative to pre-advice baseline averaged across trials for one representative participant in midbrain and DLPFC (a) or OFC and amygdala (b). The shaded areas depict + / - standard error of this mean. In this figure, the scales reflect *t*-values.

The online version of this article includes the following figure supplement(s) for figure 7:

**Figure supplement 1.** Social versus non-social weighting (*Equation 21*).

phase (i.e. stability/volatility), and (ii) a significant interaction of task phase with source of information.

By contrasting stable against volatile phases, irrespective of information source, we found that the right supramarginal gyrus, bilateral inferior occipital gyri, postcentral/precentral gyri, and the right anterior insula were more active for stable compared to volatile periods. Furthermore, an interaction between task phase and information source showed preferential activity for stable card information in the midbrain [−4,–22, −8]. Additional activations were detected in the right OFC, VLPFC, dorsomedial cingulate gyrus, and anterior cingulate sulcus/SMA (*Figure 10*; *Table 6* and *Table 7*). These regions processed stability (vs. volatility) more strongly for card than advice information.

Importantly, the regions processing stability (vs. volatility) more strongly for advice than card information also overlapped with the arbitration signal, and included the amygdala, the superior temporal sulcus, and the ventromedial PFC (*Figure 11*). Thus, model-dependent and model-independent analyses agree in localizing arbitration to frontoparietal regions in the individual domain and to ventromedial prefrontal and amygdala regions in the social domain.

**Table 4.** MNI coordinates and t-statistic of maxima of activations induced by arbitration for the individually estimated card reward probability (**Equation 20**; p<0.05, cluster-level whole-brain corrected).
Related to **Figure 8a**.

| | Hemisphere | X | Y | Z | # Voxels | t-statistic |
|---|---|---|---|---|---|---|
| $\xi_c^{(k)}$: Positive correlations | | | | | | |
| Midbrain | L | -6 | −18 | −10 | 95 | 4.94 |
| Thalamus | L | −16 | −18 | 8 | 232 | 5.10 |
| | R | 22 | −30 | 4 | 206 | 5.10 |
| Anterior insula | L | −44 | 2 | 0 | 2232 | 7.28 |
| | R | 36 | 16 | 8 | 943 | 6.23 |
| Supplementary motor area/anterior cingulate sulcus | L | -2 | -8 | 52 | 1688 | 6.29 |
| Dorsolateral PFC | R | 36 | 46 | 30 | 136 | 5.93 |
| Middle occipital gyrus | R | 12 | −86 | 6 | 237 | 11.70 |
| | L | −32 | −82 | 16 | 136 | 8.26 |
| Superior occipital gyrus | R | 28 | −78 | 30 | 343 | 11.00 |
| | L | −26 | −82 | 32 | 143 | 8.73 |
| Cerebellum | R | 18 | −48 | −18 | 21557 | 12.91 |

## Are there neural differences in the representation of social versus non-social information?

To address the question of distinct representation of social compared to non-social signatures of learning, we investigated precision-weighted predictions of social and non-social outcomes. The precision-weighted predictions consist of the two factors that enter the computation of integrated beliefs (**Equation 21**) about the outcome. The first reflects the individual card color estimates weighted by arbitration in favor of the individually sampled card probabilities (non-social weighting), whereas the second reflects the predictions of advice accuracy weighted by arbitration in favor of the advisor (social weighting). Increased effects of non-social compared to social weighting were detected in bilateral cerebellum, occipital cortices (lingual gyrus, superior occipital cortex), left

**Table 5.** MNI coordinates and t-statistic of maxima of activations induced by arbitration for the social advice (**Equation 19**; p<0.05, cluster-level whole-brain FWE corrected).
Related to **Figure 8b**.

| | Hemisphere | X | Y | Z | # Voxels | t -statistic |
|---|---|---|---|---|---|---|
| $\xi_a^{(k)}$: Positive correlations | | | | | | |
| Precuneus | R | 6 | −51 | 32 | 284 | 6.25 |
| Amygdala | R | 18 | −10 | −16 | 107 | 5.20 |
| Anterior cingulate cortex | L | -2 | 44 | −10 | 136 | 4.82 |
| Ventromedial PFC | R | 8 | 52 | 14 | 231 | 5.72 |
| Ventrolateral PFC | R | 50 | 36 | 0 | 305 | 4.93 |
| Frontopolar cortex | R | 4 | 62 | 22 | 153 | 4.59 |
| Orbitofrontal cortex | R | 28 | 26 | −16 | 126 | 5.11 |
| Middle frontal gyrus | R | 38 | 14 | 28 | 305 | 5.36 |
| Superior temporal gyrus | L | −60 | −30 | -2 | 107 | 4.90 |
| Superior temporal sulcus | R | 52 | −18 | -8 | 152 | 5.51 |
| Anterior temporoparietal junction | R | 56 | −52 | 24 | 173 | 4.18 |
| Cerebellum | L | −24 | −84 | −34 | 121 | 4.11 |

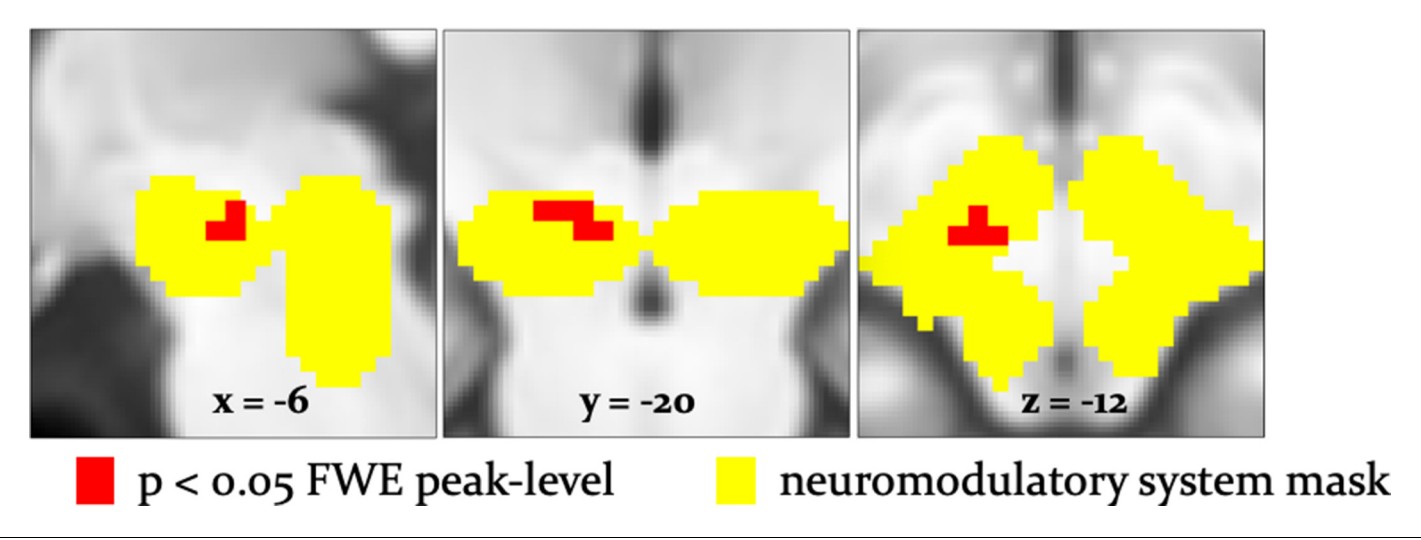

**Figure 8.** Arbitration signals in neuromodulatory ROI. Activation of the dopaminergic midbrain was associated with arbitrating in favor of individually learned information. Activation (red) is shown at p<0.05 FWE corrected for the full anatomical ROI comprising dopaminergic, cholinergic, and noradrenergic nuclei (yellow).

The online version of this article includes the following figure supplement(s) for figure 8:

**Figure supplement 1.** Neuromodulatory nuclei anatomical mask.

anterior cingulate sulcus, right supramarginal gyrus, and left postcentral gyrus. Conversely, we found increased representations of social compared to non-social weighting in the left subgenual ACC with a maximum at [−7, 36,–11] (*Figure 7—figure supplement 1*).

## Replication of hierarchical precision-weighted PE effects across learning domains

To test whether the task used in this study replicates previous findings on the representation of hierarchical precision-weighted PEs (*Diaconescu et al., 2017*; *Iglesias et al., 2013*), we performed the same model-based analysis using Bayesian surprise (equivalent to an unsigned precision-weighted

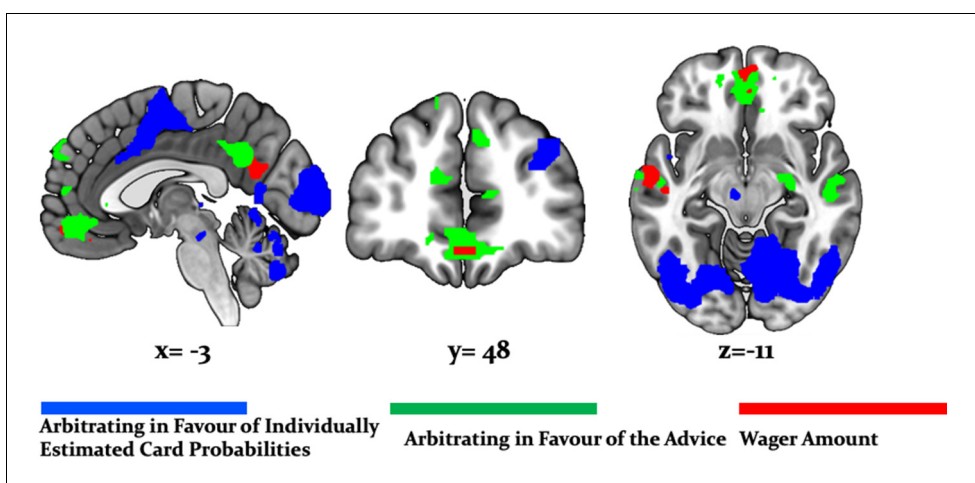

**Figure 9.** Arbitration vs. Wager Amount. Effects of arbitration (individual) (blue) were significantly larger in cortical and subcortical brain regions when compared to wager amount. Effects of arbitration in favor of social information were also significantly larger in ventromedial PFC and amygdala when compared to wager amount (green). Activity in precuneus and ventromedial PFC regions increased with increases in wager amount (magenta) (whole-brain FWE cluster-level corrected, p<0.05).

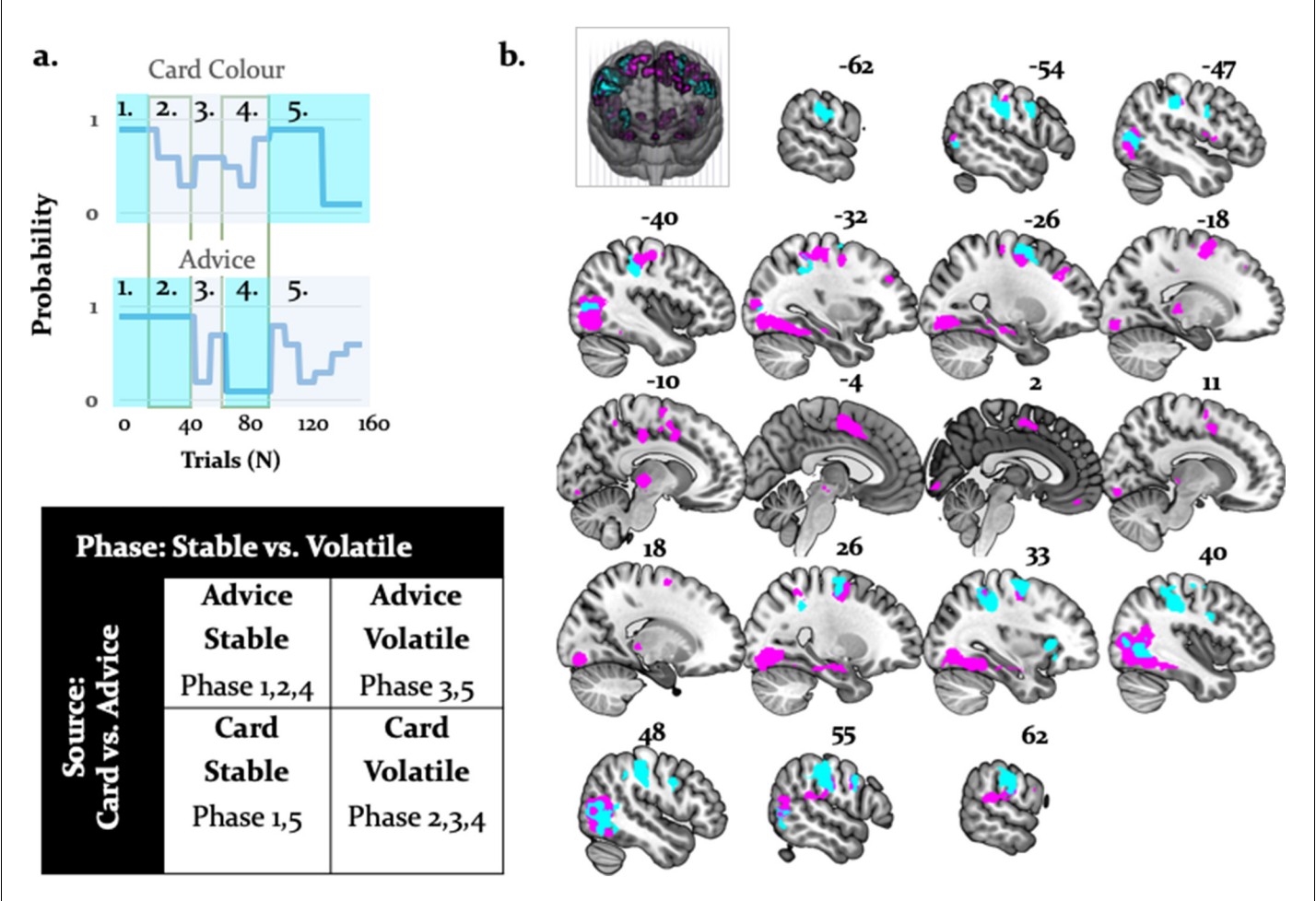

**Figure 10.** Activations related to task phase and interaction with source of information. (a) The task mapped onto a factorial structure with four conditions: (i) stable card and stable advisor, (ii) stable card and volatile advisor, (iii) volatile card and stable advisor, and (iv) volatile card and volatile advisor, as reflected by the shaded areas: blue (stable), grey (volatile). (b) The main effect of stability irrespective of source of information activated primarily parietal regions and the anterior insula (cyan, whole-brain FWE cluster-level corrected, p<0.05). Moreover, the interaction between task phase and source of information was localized to left midbrain, occipital regions, anterior insula, thalamus, middle cingulate sulcus, SMA, OFC, and VLPFC (magenta, whole-brain FWE cluster-level corrected, p<0.05).

outcome PE; the absolute value of *Equation 14*). Replicating the previous study (*Iglesias et al., 2013*), we found that the outcome-related BOLD activity of the substantia nigra positively correlated with the unsigned precision-weighted outcome PE, as did the bilateral inferior/middle occipital gyri, anterior insula, (ventro)lateral PFC, and the intraparietal sulcus (*Figure 6—figure supplement 1a* and *Supplementary file 1A*). In the previous study, participants predicted a visual outcome using an auditory cue (*Iglesias et al., 2013*). Thus, the PE coding of these regions seems to be sensory modality-independent.

With respect to the signed precision-weighted advice PE (*Equation 8*), we also replicated results from another recent study (*Diaconescu et al., 2017*) that employed a different advice-taking paradigm, where participants learned about advice and integrated it along with unambiguous individual information to predict the outcome of a binary lottery. Effects of signed precision-weighted advice PE were detected in right VTA/substantia nigra, the right insula, left middle temporal cortex, right dorsolateral, left dorsomedial and middle frontal cortex (*Figure 6—figure supplement 1b* and *Supplementary file 1B*).

Please note that we used the unsigned (absolute) precision-weighted PEs for the card outcomes, but the signed precision-weighted PEs for the advice. In the case of the card, the sign of this PE depends on an arbitrarily chosen coding of the color and the sign is meaningless (see *Iglesias et al.,*

**Table 6.** MNI coordinates and F-statistic for main effects of stability (p<0.05, FWE whole-brain corrected).
Related to *Figure 11* (activations in cyan).

| | Hemisphere | X | Y | Z | # Voxels | *F-statistic* |
|---|---|---|---|---|---|---|
| Stability > Volatility | | | | | | |
| Supramarginal gyrus | R | 46 | −28 | 42 | 1199 | 38.16 |
| Inferior occipital gyrus | R | 46 | −66 | 0 | 580 | 33.99 |
| | L | −46 | −70 | 4 | 256 | 20.82 |
| Anterior insula | R | 34 | 20 | 2 | 98 | 29.30 |
| Postcentral gyrus | L | −52 | 2 | 34 | 107 | 28.97 |
| | R | 54 | −22 | 34 | 129 | 5.59 |
| Precentral gyrus | L | −60 | −20 | 32 | 512 | 40.21 |
| | R | 50 | 4 | 32 | 129 | 20.58 |
| Middle frontal gyrus | L | −26 | 0 | 58 | 117 | 20.18 |

*2013*). In contrast, for the advice, the sign refers to the valence and instances of surprise where the advisor was more helpful than predicted, and may have a different meaning than instances of surprise where the advisor was more misleading than predicted (see *Diaconescu et al., 2017*). For completeness, we also investigated the neural correlates of the signed reward precision-weighted PE and noted a similar network of posterior parietal and dorsolateral prefrontal regions.

Effects of precision-weighted volatility PEs for card outcomes were represented in the right superior temporal gyrus, supramarginal gyrus, and posterior insula (*Figure 6—figure supplement 2a*) while the effects of precision-weighted volatility PEs for the adviser fidelity were encoded in the right anterior supplementary motor area (SMA) and anterior insula.

**Table 7.** MNI coordinates and F-statistic for interactions between task phases and stimulus type (p<0.05, FWE whole-brain corrected).
Related to *Figure 11* (activations in magenta).

| | Hemisphere | X | Y | Z | # Voxels | *F-statistic* |
|---|---|---|---|---|---|---|
| Information Source × Task Phase | | | | | | |
| Midbrain | L | -4 | −22 | -8 | 154 | 48.03 |
| Thalamus | L | −12 | −24 | 0 | 189 | 116.73 |
| | R | 16 | −30 | 2 | 154 | 104.27 |
| Middle cingulate gyrus | L | −10 | 16 | 32 | 94 | 37.10 |
| Anterior insula | L | −34 | -2 | 10 | 88 | 26.71 |
| Supplementary motor area/anterior cingulate sulcus | L | -6 | -2 | 56 | 736 | 104.45 |
| Dorsolateral PFC | L | −38 | 52 | 8 | 133 | 22.96 |
| | R | 34 | 34 | 34 | 94 | 21.02 |
| Inferior occipital gyrus | R | 44 | −66 | 6 | 3600 | 190.83 |
| | L | −40 | −76 | −12 | 3300 | 162.67 |
| Superior occipital gyrus | R | 28 | −78 | 30 | 80 | 23.54 |
| | L | −26 | −82 | 32 | 81 | 28.64 |
| Orbitofrontal cortex | L | 0 | 48 | −22 | 189 | 100.84 |
| | R | 2 | 40 | −24 | 180 | 34.66 |
| Ventrolateral prefrontal cortex | L | −46 | 48 | −12 | 81 | 37.69 |
| | R | 50 | 44 | -8 | 80 | 23.53 |
| Cerebellum | R | 30 | −86 | −42 | 95 | 25.15 |

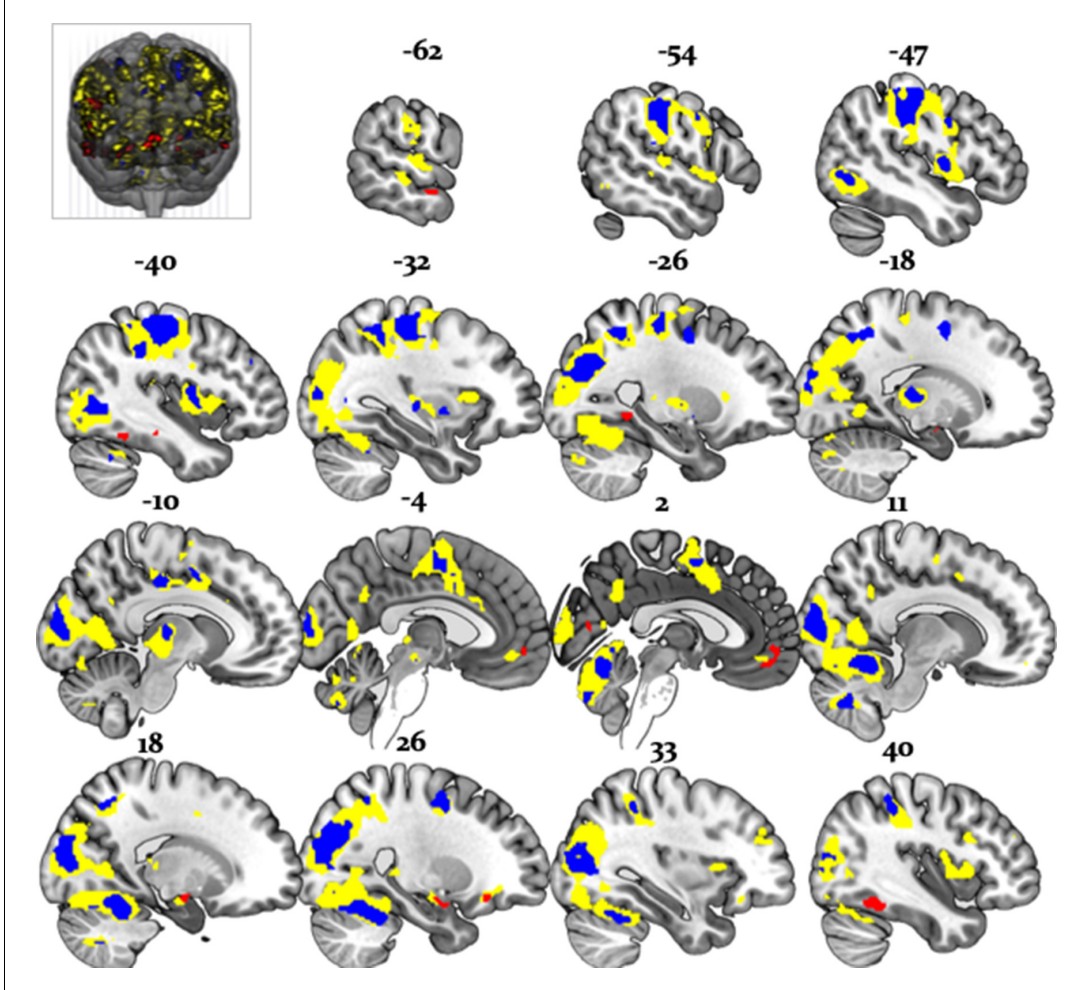

**Figure 11.** Overlap between model-dependent and model-independent results. Arbitration signal (*Equation 19*) (yellow) overlapped with the regions showing an enhanced effect of stability for individual compared to social learning systems (blue) and regions showing enhanced effects of stability in the social compared to individual learning systems (red) (whole-brain FWE peak-level corrected, p<0.05).

Finally, we also replicated the finding that higher-level, volatility PEs (*Equations 13 and 15*) were represented in cholinergic regions. This time, however, we observed effects of advice volatility precision-weighted PEs in the cholinergic nuclei in the tegmentum of the brainstem, that is, the pedunculopontine tegmental (PPT) and laterodorsal tegmental (LDT) nuclei (p<0.05 FWE voxel-level within an anatomical mask including all cholinergic nuclei) (*Figure 6—figure supplement 2b*).

## Discussion

Our study shows how healthy participants arbitrate between uncertain social and individual information under varying conditions of stability during a binary lottery task. (*Figure 1*). Participants arbitrated between the two information sources by taking into account their relative precision. The more precise one information source was over the other and the more stable the advisor was perceived to be, the more points participants were willing to wager.

By showing that participants tracked the volatility of both the advice and the card color probabilities (*Figure 3*), our study underscores the importance of volatility in arbitrating between social advice and individual reward-relevant information. At the behavioral level, trial-by-trial accuracy of participant predictions, frequency of taking advice into account, and amount of points wagered on each trial (*Figure 5—figure supplement 1*) were all reduced by volatility. Thus, in stable compared to volatile environments, the propensity for arbitration in favor of the more precise information

source increases. Numerous studies have demonstrated an important role of volatility in higher level learning (*Behrens et al., 2007*; *Behrens et al., 2008*; *Nassar et al., 2010*; *Iglesias et al., 2013*; *Vossel et al., 2014*; *Diaconescu et al., 2017*; *Pulcu and Browning, 2017*), in-keeping with the present findings.

## Evidence for domain-generality of arbitration in lateral prefrontal cortex

Using both model-based and model-independent (factorial) fMRI analysis, we found that the arbitration signal correlated with activity in dorsolateral and ventrolateral PFC, frontopolar, and orbitofrontal cortex (*Figures 6* and *11*). These findings corroborate previous insights on arbitration between different forms of individual information also pointing to lateral prefrontal cortex (*Lee et al., 2014*), in line with domain generality for arbitrating. Note though that arbitration activity in the prefrontal cortex followed a self-versus-other axis: dorsal prefrontal activity increased the more strongly participants weighed their own predictions of reward probabilities over the perceived reliability of the advisor. Conversely, activity in the ventromedial PFC and orbitofrontal cortex showed the opposite pattern and increased in activity as participants relied more heavily on their own reward probability estimates relative to the advice (*Figure 7*). Together, arbitration appears to be sensitive to the source of information entering the arbitration process, contrary to an entirely domain-general process.

## Arbitration in the dopaminergic system

The results of both model-based and factorial analyses suggest a key role of the midbrain in arbitrating for individual estimates about card color over advice (*Figure 8*). Primate studies found that sustained dopamine neuron activity signaled expected uncertainty (*Fiorillo et al., 2003*; *Schultz, 2010*; *Schultz et al., 2008*). This was further supported by human pharmacological studies (*Burke et al., 2018*; *Ojala et al., 2018*) as well as fMRI research showing possible involvement of dopamine in risk taking and of dopaminoceptive regions, such as the caudate, anterior insula, ACC and the medial PFC in uncertainty coding (e.g. *Dreher et al., 2006*; *Preuschoff et al., 2008*; *Tobler et al., 2009*) and social advice predictions under uncertainty (*Henco et al., 2020*). In particular, studies employing hierarchical Bayesian models have identified ventral tegmental area/substantia nigra activation correlated to precision of predictions about desired outcomes (*Friston et al., 2014*; *Schwartenbeck et al., 2015*).

These findings may also underscore the role of dopamine in modulating participants' ability to optimize learning to suit ongoing estimates of environmental volatility. Potential neurobiological mechanisms include meta-learning models, which propose an important role of phasic dopamine signals in training prefrontal system dynamics, to infer on the statistical structure of the environment (*Collins and Frank, 2016*; *Wang et al., 2018*). Such models imply that improved learning of the structure of the environment, for example current levels of volatility, results in more appropriate arbitration adjustment.

## Arbitrating in favor of the advisor activates the amygdala and orbitofrontal cortex

The amygdala processed perceived reliability of social information, reflected in activity increasing the more participants discounted their own estimates of rewarded card color probabilities in favour of the advisor's recommendations. The amygdala has been implicated in processing facial expressions related to affective ToM (*Schmitgen et al., 2016*) and more generally, processing affective value and motivational significance of various stimuli, including other people (*Güroğlu et al., 2008*; *Zink et al., 2008*; *Zerubavel et al., 2015*). Together these findings suggest that the amygdala may represent the uncertainty of socially-relevant stimuli, inferred from processing the intentions of others.

Similar to the amygdala, the orbitofrontal cortex showed a significant interaction between task phase and information source, indicative of arbitrating in favor of social information. This finding is consistent with the hypothesis that the orbitofrontal cortex and other areas of the social brain evolved to enable primates and particularly humans to successfully navigate complex social situations (*Dunbar, 2009*). This notion received support from strong positive correlations between

orbitofrontal cortex grey matter volume and social network size (*Powell et al., 2012*), as well as socio-ocognitive abilities (*Powell et al., 2010*; *Scheuerecker et al., 2010*). Furthermore, in-keeping with a role of orbitofrontal cortex in mental state attribution for ambiguous social stimuli (*Deuse et al., 2016*), our findings suggest that this region reduces the uncertainty of social cues that signal changes in intentionality.

With respect to social learning signatures, we observed that the sulcus of the ACC represents predictions related to one's own estimates of the card color outcomes, whereas the subgenual ACC represents predictions about the advisor's fidelity. This is consistent with previous findings that the sulcus of ACC dorsal to the gyrus plays a domain-general role in motivation (*Rushworth et al., 2007*; *Rushworth and Behrens, 2008*; *Apps et al., 2016*), whereas the gyrus of the ACC signals information related to other people (*Behrens et al., 2008*; *Apps et al., 2013*; *Apps et al., 2016*; *Lockwood, 2016*).

## Implications for mentalizing disorders

An intriguing extension of the current study concerns the question of whether arbitration occurs differently in patients with psychiatric and neurodevelopmental disorders involving ToM processes. If so, how do these processing differences affect behavior? For example, individuals with autism spectrum disorder may preferentially rely on their own experiences rather than on the recommendations of others. Indeed, they appear to represent social prediction errors less strongly than individuals without autism (*Balsters et al., 2017*). Accordingly, they may be able to better infer the volatility of the card color probability compared to the advice in our task. In contrast, patients with schizophrenia may be overly confident about their ability to judge advice validity due to fixed beliefs about the advisor's intentions (*Freeman and Garety, 2014*) or show an over-reliance on social information in line with accounts of over-mentalization in this disorder (*Montag et al., 2011*; *Andreou et al., 2015*). Future work may test these intriguing possibilities.

## Limitations

One limitation of our study is that it did not include reciprocal social interactions, but rather used pre-recorded videos of human partners. ToM processes may be more prominent in interactive paradigms (*Diaconescu et al., 2014*) or interactions that involve higher levels of recursive thinking (*Devaine et al., 2014a*; *Devaine et al., 2014b*). By extension, our study may have limited generalizability to real-world social interactions. However, assessing arbitration between social and individual information necessitated the standardization of the advice given to each participant. To make the task as close as possible to a realistic social exchange, the videos of the advisor were extracted from trials when they truly intended to help or truly intended to mislead. More importantly, to adequately compare learning from social and individual information in stable and volatile phases, we needed to ensure that the two information types were orthogonal to each other and balanced in terms of volatility.

Second, we did not include a non-social control task. Thus, it is unclear how 'social' the presently investigated form of learning about the advisor's fidelity and volatility actually is. The differences in activated regions at least suggest that our participants processed the two sources of information differently. However, whether the process we identified is specifically social in nature or rather reflects learning from an indirect information source needs to be examined in future studies by including an additional control condition.

In order to distinguish general inference processes under volatility from inference specific to intentionality, we previously included a control task (*Diaconescu et al., 2014*), in which the advisor was blindfolded and provided advice with cards from predefined decks that were probabilistically congruent to the actual card color. This control task closely resembled the main task, with the exception of the role of intentionality. Model selection results suggested that participants in the control task did not incorporate time-varying estimates of volatility about the advisor into their decisions. In the current study, we tested this by including models without volatility, but found that they performed substantially worse than models with volatility (see *Figure 2* and *Table 2a* for details). Thus, our participants appeared to process advisor intentionality.

## Conclusions

Our study indicates that arbitrating between social and individual sources of information corresponds to weighing the relative reliability of each source. This process appears to engage different brain regions for social and individual information, in-keeping with domain specificity. However, the lateral prefrontal cortex appears to adjudicate between several different types of learning, in-keeping with domain generality. These findings contribute to our understanding of arbitration in neuro-typical individuals, which may provide a knowledge basis for future insight into disorders with impaired arbitration.

# Materials and methods

## Participants

We recruited 48 volunteers (mean age 23.6 ± 1.4, 32 females) who were non-smokers, right-handed, and had normal or corrected-to-normal vision. Participants had no history of neurological or psychiatric illness, or of drug abuse. Psychology students were excluded from participation because of previous exposure to similar advice-taking paradigms in their courses. Participants were asked to abstain from alcohol 24 hr prior to the study and from medication, including aspirin, 3 days prior to the study. We did not analyse the data of 10 participants: two pilot participants; one participant who stopped the experiment midway due to head pain; one participant who fell asleep; and six participants where stimulus presentation malfunctioned during the experiment. Altogether, 38 participants (mean age 24.2 ± 1.3; 26 females) entered the final analysis.

## Stimuli and task

We modified the deception-free binary lottery game of *Diaconescu et al., 2014*. In each trial, the participant had to predict the color of a card draw – blue or green. Participants could base their predictions on social information and/or on individually experienced recent outcome history (see below). They received social information from the 'advisor', who held up a card in one of the two colors before every draw, recommending to the participant which option to choose. The advisor based his or her suggestion on information that was true with a probability of 80%, although the participants were not informed of this fact. Furthermore, the advisor received monetary incentives to change his or her strategy and thus provide either helpful or misleading advice at different stages of the game (*Figure 1b*) with the average probability of advice being correct in 56% of trials. To compare participants in terms of their learning and decision-making parameters, we needed to standardize the advice. This means that each participant received the same input sequence, that is order and type of videos.

To display social information in a standardized fashion and gender-match advisors and participants, we created videos from two male and two female advisors, who changed their advice as a function of the incentives in a previously recorded face-to-face session (see *Diaconescu et al., 2014*). Their advice on each trial was recorded for an entire experimental session and the full-length videos were edited into 2 s segments, focusing on the advice period. We received informed consent from all advisors in the initial (face-to-face) behavioral study to record and use the advice-giving videos in subsequent studies. All video clips were matched in terms of their luminance, contrast, and color balance using Adobe Photoshop Premiere CS6.

To standardize the advice, avoid implicit cues of deception, and make the task as close as possible to a social exchange in real time, the videos of the advisor were extracted from trials when they truly intended to help or truly intended to mislead. Although each participant received the same advice sequence throughout the task, the advisors displayed in the videos varied between participants, in order to ensure that physical appearance and gender did not impact on their decisions to take advice into account. Advisor-to-participant assignment was randomized (within the gender-matching constraint) and balanced. We found no differences in performance and degree of reliance on advice between the four advisors: $F(1,36) = 1.82$, p=0.16.

In contrast to previous studies (*Diaconescu et al., 2014*; *Diaconescu et al., 2017*), participants had to infer card color probabilities (blue versus green) from individually experienced outcomes of previous trials rather than being provided with (changing) pie charts explicitly stating the probabilities. In each trial, they had to arbitrate between following either social information (previous advice,

inferring on intention) or individual information (previous cards, inferring on probability). Moreover, also in contrast to previous studies, for each lottery prediction, participants wagered between one and ten points to indicate how confident they were about their predictions. The tick mark on the wager bar was randomly positioned in each trial to avoid providing a reference point (a regression analysis confirmed that the starting position of the wager indeed failed to explain each participant's trial-wise wager selection, $t(37) = -0.89$, p=0.31). Depending on the correctness of the prediction, the wager was added to or subtracted from the cumulative score and thereby affected the participant's payment at the end of the experiment (see below).

Each trial (*Figure 1a*) began with a video of the advisor holding up a card, followed by a decision screen in which participants selected the blue or green card. At the next screen, they were asked to provide the wager. The subsequent outcome screen revealed the drawn card. Finally, the updated cumulative score appeared. The color-to-button assignment used to convey the lottery prediction (blue or green) and the orientation of the wager bar were randomized between participants to prevent confounding with visuomotor processes.

Across trials, the color-reward probabilities and the advisor intentions varied independently of each other. In other words, the probability distributions of the two information sources – card color and advice – were designed to be statistically independent. This allowed for a $2 \times 2$ factorial design structure, where trials could be divided into four conditions: (i) stable card and stable advisor, (ii) stable card and volatile advisor, (iii) volatile card and stable advisor, and (iv) volatile card and volatile advisor in a total of 160 trials (*Figure 1b*). Based on this factorial structure, we predicted that arbitration signals would vary as a function of the stability of each information source.

## Procedure

We explained the deception-free task to participants and ensured their comprehension with a written questionnaire, which required them to describe the instructions in their own words. The task instructions, which were originally presented to participants in their native German, were translated into English for the purpose of this paper. Pronouns were adapted to the advisor's gender: "The advisor has generally more information than you about the outcome on each trial. The objective of the advisor is to use this information to guide your choices and reach his/her own goals. Note that the advisor does not have 100% accurate information about which color 'wins' and he/she might be incorrect. Nevertheless, he/she will on average have better information than you and his/her advice may be valuable to you." The actual experiment was divided into two sessions, with a 2-min break in the middle when participants could close their eyes and rest. The first session included 70 trials and the second session 90 trials.

To test the construct validity of our computational model and verify whether participants inferred on the advisor's fidelity, we asked them to rate the usefulness of the advisor's card recommendation based on a multiple choice question (including, 'helpful,' 'misleading,' or 'neutral'). This question was presented six times throughout the task and responses allowed us to assess whether at any point in time, the model could significantly predict participants' responses.

Participants could earn a bonus of 10 Swiss Francs for a cumulative score of at least 380 points, and a bonus of 20 Swiss Francs for winning more than 600 points. Importantly, participants were not given any information about the bonus thresholds in order to prevent induction of local risk-seeking or risk-averse wagering behavior (reference point effects) when participants were close to a threshold. Participants on average reached the first reward bonus and were paid $82.3 \pm 8.4$ Swiss Francs (including the performance-dependent bonus) at the end of the study. After the task, participants completed a debriefing questionnaire, and we revealed to them the general trajectory of the advisor's intentions.

## Data acquisition and preprocessing

We acquired functional magnetic resonance images (fMRI) from a Philips Achieva 3T whole-body scanner with an 8-channel SENSE head coil (Philips Medical Systems, Best, The Netherlands) at the Laboratory for Social Neural Systems Research at the University Hospital Zurich. The task was presented on a display at the back of the scanner, which participants viewed using a mirror placed on top of the head coil. The first five volumes of each session were discarded to allow for magnetic saturation.

During the task, we acquired gradient echo T2*-weighted echo-planar imaging (EPI) data with blood-oxygen-level dependent (BOLD) contrast (slices/volume = 33; TR = 2665 ms; voxel volume = 2×2 x 3 mm$^3$; interslice gap = 0.6 mm; field of view (FOV) = 192×192 x 120 mm; echo time (TE) = 35 ms; flip angle = 90°). The images were oblique, slices with −20° right-left angulation from a transverse orientation. The entire experiment comprised 1300 volumes, with 600 volumes in the first session and 700 in the second. Heart rate and breathing of the participants were recorded for physiological noise correction purposes using ECG and a pneumatic belt, respectively.

We also measured the homogeneity of the magnetic field with a T1-weighted 3-dimensional (3-D) fast gradient echo sequence (FOV = 192×192 x 135 mm$^3$; voxel volume = 2×2 x 3 mm$^3$; flip angle = 6°; TR = 8.3 ms; TE1 = 2 ms; TE2 = 4.3 ms). After the experiment, we acquired T1-weighted structural scans from each participant using an inversion-recovery sagittal 3-D fast gradient echo sequence (FOV = 256×256 x 181 mm$^3$; voxel volume = 1×1 x 1 mm$^3$; TR = 8.3 ms; TE = 3.9 ms; flip angle = 8°).

The software package SPM12 version 6470 (Wellcome Trust Centre for Neuroimaging, London, UK; http://www.fil.ion.ucl.ac.uk/spm) was used to analyse the fMRI data. Temporal and spatial pre-processing included slice-timing correction, realignment to the mean image, and co-registration to the participant's own structural scan. The structural image underwent a unified segmentation procedure combining segmentation, bias correction, and spatial normalization (*Ashburner and Friston, 2005*); the same normalization parameters were then applied to the EPI images. As a final step, EPI images were smoothed with an isotropic Gaussian kernel of 6 mm full-width half-maximum.

BOLD signal fluctuations due to physiological noise were modeled with the PhysIO toolbox (http://www.translationalneuromodeling.org/tapas) (*Kasper et al., 2017*) using Fourier expansions of different order for the estimated phases of cardiac pulsation (3rd order), respiration (4th order) and cardio-respiratory interactions (1st order; *Glover et al., 2000*). The 18 modeled physiological regressors entering the subject-level GLM along with the six rigid-body realignment parameters and regressors of interest were used to account for BOLD signal fluctuations induced by cardiac pulsation, respiration, and the interaction between the two.

## Computational modeling

We formalized arbitration in terms of hierarchical Bayesian inference as the relative perceived reliability of each information source. In other words, arbitration was defined as a ratio of precisions: the precision of the prediction about advice accuracy and color probability, divided by the total precision. The precisions of the predictions afforded by each learning system are obtained by applying a two-branch hierarchical Gaussian filter (*Mathys et al., 2011*; *Mathys et al., 2014*) along with a response model (see below) to participants' trial–wise behavior (i.e. choices and wagers).

## Learning model: Hierarchical Gaussian Filter

The HGF is a model of hierarchical Bayesian inference widely used for computational analyses of behavior (e.g. [*Iglesias et al., 2013*; *Vossel et al., 2014*; *Hauser et al., 2014*; *de Berker et al., 2016*; *Marshall et al., 2016*]). To apply it to our task, we assumed that the rewarded card color (individual learning) and the advice accuracy (social learning) varied as a function of hierarchically coupled hidden states: $x_1^{(k)}, x_2^{(k)}, \ldots, x_n^{(k)}$. They evolved in time by performing Gaussian random walks. At every level, the step size was controlled by the state of the next-higher level (*Figure 2a*).

Starting from the bottom of the hierarchy, states $x_{1,\,a}$ and $x_{1,c}$ represented binary variables, namely the advice accuracy (1 for accurate, 0 for inaccurate) and the rewarded card color (1 for blue, 0 for green). All states higher than $x_1$ were continuous. They denoted (i) the advisor fidelity and tendency for a given card color to be rewarded, and (ii) the rate of change of the advisor's intentions and card color contingencies, respectively. Four learning parameters, namely, $\kappa_a$, $\kappa_c$, $\vartheta_a$ and $\vartheta_c$ determined how quickly the hidden states evolved in time. Parameter $\kappa$ represented the degree of coupling between the second and the third levels in the hierarchy, whereas $\vartheta$ determined the variability of the volatility over time (meta-volatility). This constitutes the *generative model* of the process producing the outcomes observed by participants. The overall model and the formal equations describing these relations in a social learning context are detailed in *Diaconescu et al., 2014*.

## Model inversion: agent-specific arbitration

In accordance with Bayes' rule, we assumed that participants who make inferences on advice and card colors form posterior beliefs over the hidden states (i.e. congruency of advice with actual card color; rewarded card color) based on the outcomes they observe. Model inversion is the application of Bayes' rule to a generative model such as the one described above. This leads to a *recognition* or *perceptual model*, which describes participants' beliefs about hidden states. Assuming Gaussian distributions, these agent-specific beliefs are denoted by their summary statistics, that is μ (mean) and $\sigma$ (variance/uncertainty) or the inverse of the variance $\pi = 1/\sigma$ (precision/certainty).

Using variational Bayes under the mean-field approximation, simple analytical trial-by-trial update equations can be derived. The posterior means $\mu_i^{(k)}$ or predictions on each trial $k$ at each level of the hierarchy $i$ change as a function of precision-weighted prediction errors (PEs):

$$\Delta\mu_i^{(k)} \propto \frac{\hat{\pi}_{i-1}^{(k)}}{\pi_i^{(k)}} \delta_{i-1}^{(k)} \tag{1}$$

Throughout, predictions or prior beliefs about the hidden states (before observing the outcome) are denoted with a hat symbol. States $\hat{\pi}_{i-1}^{(k)}$ and $\pi_i^{(k)}$ represent the estimated precisions about (i) the input from the level below (i.e. precision of the data – advice congruency or rewarded card color) and (ii) the belief at the current level, respectively.

The updates about the advisor's fidelity are:

$$\Delta\mu_{2,a}^{(k)} = \frac{1}{\pi_{2,a}^{(k)}}\delta_{1,a}^{(k)} \tag{2}$$

where

$$\delta_{1,a}^{(k)} = u^{(k)} - \hat{\mu}_{1,a}^{(k)}. \tag{3}$$

Variable $u^{(k)}$ is the sensory input at trial $k$, where given advice is either accurate $\left(u^{(k)} = 1\right)$ or inaccurate $\left(u^{(k)} = 0\right)$. Furthermore, $\hat{\mu}_{1,a}^{(k)}$ corresponds to the logistic sigmoid of the current expectation of the advisor fidelity:

$$\hat{\mu}_{1,a}^{(k)} = s\left(\mu_{2,a}^{(k-1)}\right) = \frac{1}{1 + \exp\left(-\mu_{2,a}^{(k-1)}\right)} \tag{4}$$

The current belief precision is equivalent to:

$$\pi_{2,a}^{(k)} = \hat{\pi}_{2,a}^{(k)} + \frac{1}{\hat{\pi}_{1,a}^{(k)}} \tag{5}$$

with the predicted (i) belief precision $\hat{\pi}_{2,a}^{(k)}$ and (ii) the sensory, lower-level precision about the advice $\hat{\pi}_{1,a}^{(k)}$ computed as:

$$\hat{\pi}_{2,a}^{(k)} = \frac{1}{\frac{1}{\pi_{2,a}^{(k-1)}} + \exp\left(\kappa\mu_{3,a}^{(k-1)} + \omega\right)} \tag{6}$$

$$\hat{\pi}_{1,a}^{(k)} = \frac{1}{\hat{\mu}_{1,a}^{(k)}\left(1 - \hat{\mu}_{1,a}^{(k)}\right)}. \tag{7}$$

Thus, the advice belief precision depends on (i) the predicted sensory precision of the input $\hat{\pi}_1^{(k)}$, and (ii) the predicted volatility, $\mu_{3,a}^{(k-1)}$ from the level above via *Equation 6*.

The precision-weighted PE about the advice, which is used to update the belief about fidelity is equivalent to:

$$\varepsilon_{2,a} = \frac{1}{\pi_{2,a}^{(k)}} \delta_{1,a}^{(k)} \tag{8}$$

Going up the hierarchy, the updates of advice volatility are proportional to precision-weighted PEs:

$$\Delta \mu_{3,a}^{(k)} \propto \frac{1}{\pi_{3,a}^{(k)}} \delta_{2,a}^{(k)}. \tag{9}$$

They depend on the higher-level volatility PE $\delta_{2,a}$:

$$\delta_{2,a}^{(k)} = \frac{\hat{\pi}_{2,a}^{(k)}}{\pi_{2,a}^{(k)}} + \left( \pi_{2,a}^{(k)} \right)^2 \hat{\pi}_{2,a}^{(k)} \left( \Delta \mu_{2,a}^{(k)} \right)^2 - 1, \tag{10}$$

and the higher level volatility precision $\pi_3$:

$$\pi_{3,a}^{(k)} = \hat{\pi}_{3,a}^{(k)} + \frac{1}{2} \left( \gamma_{2,a}^{(k)} \right)^2 + \left( \gamma_{2,a}^{(k)} \right)^2 \delta_{2,a}^{(k)} - \frac{1}{2} \gamma_{2,a}^{(k)} \delta_{2,a}^{(k)}, \tag{11}$$

with the precision of the prediction about volatility given by

$$\hat{\pi}_{3,a}^{(k)} = \frac{1}{\frac{1}{\pi_{3,a}^{(k-1)}} + \vartheta_a}. \tag{12}$$

The third level, the precision-weighted volatility PE is equivalent to:

$$\varepsilon_{3,a} = \frac{1}{\pi_{3,a}^{(k)}} \delta_{2,a}^{(k)}. \tag{13}$$

The same form of update equations (and precision-weighted PEs) can be derived for the individual information source, updating beliefs about the rewarded card color, i.e.:

$$\varepsilon_{2,c} = \frac{1}{\pi_{2,c}^{(k)}} \delta_{1,c}^{(k)} \tag{14}$$

and

$$\varepsilon_{3,c} = \frac{1}{\pi_{3,c}^{(k)}} \delta_{2,c}^{(k)}. \tag{15}$$

The prediction errors exhibit a similar form as for the advice, with

$$\delta_{1,c}^{(k)} = u^{(k)} - \hat{\mu}_{1,c}^{(k)} \tag{16}$$

for the outcome PE and

$$\delta_{2,c}^{(k)} = \frac{\hat{\pi}_{2,c}^{(k)}}{\pi_{2,c}^{(k)}} + \left( \pi_{2,c}^{(k)} \right)^2 \hat{\pi}_{2,c}^{(k)} \left( \Delta \mu_{2,c}^{(k)} \right)^2 - 1 \tag{17}$$

for the card volatility PE. The individually estimated card color probability is equivalent to the logistic sigmoid of the current expectation of the rewarding card color:

$$\hat{\mu}_{1,c}^{(k)} = s \left( \mu_{2,c}^{(k-1)} \right) = \frac{1}{1 + exp \left( -\mu_{2,c}^{(k-1)} \right)}. \tag{18}$$

In this context, Bayes-optimality is individualized with respect to the values of the learning parameters, which were allowed to differ across participants.

## Arbitration signal

Within this computational framework, we defined arbitration as the relative perceived precision associated with each information source, which is equivalent to the precision of the prediction of each information channel (advice or card; i.e. $\hat{\pi}$) divided by the total precision. Arbitration is consistent with Bayes' rule representing the optimal integration of the two inferred states by their precisions.

Arbitration toward advice – that is the perceived reliability of the social information source is equivalent to:

$$\xi_{i,a}^{(k)} = \frac{\zeta \hat{\pi}_{i,a}^{(k)}}{\zeta \hat{\pi}_{i,a}^{(k)} + \hat{\pi}_{i,c}^{(k)}} \tag{19}$$

on each trial $k$ at each level of the hierarchy $i$ with $\zeta$ as the social bias or the additional bias towards the advice.

At the first level and at $i = 1$, the participant relies preferentially on the social input during action selection when $\xi_{1,a}^{(k)}$ exceeds 0.5. Conversely, when $\xi_{1,a}^{(k)}$ is below 0.5 , the participant relies more on individual (estimates of) card color probabilities:

$$\xi_{1,c}^{(k)} = \frac{\hat{\pi}_{1,c}^{(k)}}{\zeta \hat{\pi}_{1,a}^{(k)} + \hat{\pi}_{1,c}^{(k)}} = 1 - \xi_{1,a}^{(k)} \tag{20}$$

## Response model

To map beliefs to decisions, we assumed that the prediction of card color on a given trial $k$ is a function of arbitration and of the predictions afforded by each source (see **Equation 21**). The response model predicts two components of the behavioral response: (i) the participant's decision to accept or reject the advice and (ii) the number of points wagered on every trial. Responses were coded as $y = 1$ when participants took the advice and chose the card color indicated by the advisor, and $y = 0$ when participants decided against following the advice and chose the opposite card color. The expected outcome probability is thus a precision-weighted sum of the two information sources, the estimates of advice accuracy and rewarding color probability.

$$\mu_{1,b}^{(k)} = \xi_{i,a}^{(k)} \cdot \hat{\mu}_{1,a}^{(k)} + \xi_{1,c}^{(k)} \cdot \hat{\mu}_{1,c}^{(k)} \tag{21}$$

where $\xi_{i,a}^{(k)}$ and $\xi_{1,c}^{(k)}$ are the arbitration for each information source; $\hat{\mu}_{1,a}^{(k)}$ is the expected advice accuracy (**Equation 4**) and $\mu_{1,c}^{(k)}$ is the transformed expected card color probability from the perspective of the advice (i.e. the estimated card color probability indicated by the advisor).

It follows from **Equation 21**, that social weighting is represented by the first term of this integrated sum – that is $\xi_{i,a}^{(k)} \cdot \hat{\mu}_{1,a}^{(k)}$ whereas card color weighting is represented by the second term or $\xi_{1,c}^{(k)} \cdot \hat{\mu}_{1,c}^{(k)}$.

The probability that participants chose a particular card color according to their expectations about the outcome (**Equation 21**) was modeled by a softmax function:

$$p\left(y_{choice}^{(k)} = 1 | \hat{\mu}_{1,b}^{(k)}\right) = \frac{\hat{\mu}_{1,b}^{(k)^{\beta_{choice}}}}{\hat{\mu}_{1,b}^{(k)^{\beta_{choice}}} + (1 - \hat{\mu}_{1,b}^{(k)})^{\beta_{choice}}} \tag{22}$$

where $\beta_{choice} > 0$ is the participant-specific inverse decision temperature parameter. A low decision temperature (high $\beta_{choice}$) means always choosing the highest probability color, whereas a high decision temperature (low $\beta_{choice}$) means sampling randomly from a uniform distribution.

The number of points wagered provided us with a behavioral readout of decision confidence. We aimed to formally explain trial-wise wager responses as a linear function of various sources of uncertainty and precision associated with the lottery outcome prediction: (i) irreducible decision uncertainty or $\hat{\sigma}_b^{(k)}$ about the outcome, (ii) arbitration, (iii) informational uncertainty about the card color or the advice, and (iv) environmental uncertainty/volatility about the card color or the advice. We transformed these computational quantities down to the first level in the hierarchy using the sigmoid

transformation and used them to predict the trial-by-trial wager (*Figure 5* for the group average of each of these quantities):

$$\log\left(y_{wager}\right) = \beta_0 + \beta_1 \hat{\sigma}_b^{(k)} + \beta_2 \xi_1^{(k)} + \beta_3 I_{2,a}^{(k)} + \beta_4 I_{2,c}^{(k)} + \beta_5 V_{3,a}^{(k)} + \beta_6 V_{3,c}^{(k)} \tag{23}$$

with

$$\hat{\sigma}_b^{(k)} = \hat{\mu}_{1,b}^{(k)}\left(1 - \hat{\mu}_{1,b}^{(k)}\right). \tag{24}$$

Parameter $\zeta$ captures the social bias in arbitration (*equation 19*) and $I_{2,a}^{(k)}$ is the informational uncertainty about the advisor fidelity

$$I_{2,a}^{(k)} = \hat{\mu}_{1,a}^{(k)}\left(1 - \hat{\mu}_{1,a}^{(k)}\right)\hat{\sigma}_{2,a}^{(k)} \tag{25}$$

where $\hat{\sigma}_{2,a}^{(k)}$ is the inverse of $\hat{\pi}_{2,a}^{(k)}$ and represents the informational uncertainty of the prediction about the advisor's fidelity (*Equation 6*).

The environmental volatility is defined as:

$$V_{3,a}^{(k)} = \hat{\mu}_{1,a}^{(k)}\left(1 - \hat{\mu}_{1,a}^{(k)}\right)\exp\left(\mu_{3,a}^{(k-1)}\right). \tag{26}$$

Equivalent equations can be derived for the individual information source.

The trial-wise wager amount predicted by the model is then defined as:

$$\hat{y}_{wager} \overset{\text{def}}{=} \log\left(y_{wager}\right) + \sqrt{\beta_{wager}} \tag{27}$$

where $\beta_{wager}$ is a stochasticity parameter associated with the wager amount. For the priors of all $\beta$ parameters estimated here, please refer to *Table 2*.

## Competing models

To contrast competing mechanisms underlying learning and arbitration, our model space consisted of a total of 9 models (*Figure 3*). On the one hand, we included non-normative perceptual models varying in the degree of volatility processing (three-level full HGF vs. two-level no-volatility HGF) and normative perceptual models assuming optimal Bayesian inference (normative HGF). On the other hand, we included response models varying in the level of arbitration (arbitration; no arbitration: advice only; no arbitration: card information only).

We considered three families of perceptual models. The first family included the full, three-level version of the HGF (as described above). By contrast, the second family lacked the third level, and assumed that agents do not estimate the volatility of the card probabilities or the advice. Thus, comparing families with and without volatility tested whether volatility mattered for arbitrated behavior. Finally, the third family assumed a Bayes-optimal, normative process of learning from the advice and card outcomes.

In terms of response models, we also considered three families, capturing different ways in which participants may arbitrate between social and individual sources of information to make decisions. These included: (i) an 'Arbitrated' model, which assumed that participants combine and arbitrate between the two information sources, possibly unequally, (ii) an 'Advice only' model, assuming arbitration-free reliance on social information only, and (iv) a 'Card only' model, representing arbitration-free reliance on the inferred card color probabilities only (*Figure 3a*).

All models were compared formally using Bayesian model selection (BMS *Stephan et al., 2009*). Random effects BMS results in a posterior probability for each model given the participants' data. The relative goodness of models is denoted by the 'protected exceedance probability' reflecting how likely it is that a given model has a higher posterior probability than any other model in the set of models considered (*Stephan et al., 2009*; *Rigoux et al., 2014*).

We adopted a similar set of priors over the perceptual model parameters as in our previous studies (*Diaconescu et al., 2014*) (see *Table 2*). Maximum-a-posteriori (MAP) estimates of model parameters were obtained using the HGF toolbox version 3.0, freely available as part of the open source software package TAPAS at http://www.translationalneuromodeling.org/tapas.

## FMRI data analysis

### Single-subject level

Our fMRI data analysis focused on the neural mechanisms of arbitration. Specifically, we conducted two types of analyses on the pre-processed fMRI data:

First, we performed a model-based fMRI analysis, in which we constructed a general linear model (GLM), which sought to explain the high-pass filtered voxel time-series with several parametric modulators. The parametric modulators are listed below and were derived from the winning model (i.e. arbitrated three-level version of the HGF, which had the highest posterior probability at the group level). The GLMs were individualized, as the regressors were obtained from fitting the model to the behavioral data of each of the 38 participants. We individualized GLMs because participants differed in how much they relied on each information source and in the extent to which volatility influenced their trial-by-trial wagers (*Figures 4–5*). To investigate the unique contribution of each parametric modulator, we did not orthogonalize them (see *Figure 1—figure supplement 2* for correlations between them). Moreover, we also included movement and the physiological noise regressors obtained from the PhysIO toolbox (*Kasper et al., 2017*) based on ECG and respiration recordings as regressors of no interest.

In addition to arbitration at the time of advice presentation, we modeled the wager and the outcome phases to examine the effects of hierarchical precision-weighted PEs, and thus test the validity of the computational model and the reproducibility of previous findings, see *Figure 6—figure supplements 1–2* (*Iglesias et al., 2013*; *Diaconescu et al., 2017*). Specifically, the following regressors were included in the GLM:

1. Social information – time when the advice was presented (regressor duration two seconds);
2. Arbitration – parametric modulator of (1), using the trial-specific arbitration quantity (*Equation 19-20*);
3. Social Weighting – parametric modulator of (1), using the precision-weighted prediction of the advisor fidelity (first term of *Equation 21*);
4. Non-social Weighting – parametric modulator of (1), using the precision-weighted prediction of the individual card weighting (second term of *Equation 21*);
5. Wager presentation – time when the option to wager was presented (regressor duration zero seconds);
6. Wager - parametric modulator of (3), using the trial-specific amount of points wagered;
7. Outcome – time when the winning card color was presented (regressor duration zero seconds);
8. Advice Precision-weighted PE – parametric modulator of (5), using the trial-specific precision-weighted PE of advice validity (*Equation 8*);
9. Outcome Precision-weighted PE – parametric modulator of (5), using the trial-specific precision-weighted PE arising from comparing actual and predicted card color (*Equation 14*).
10. Volatility Advisor Precision-weighted PE – parametric modulator of (5), using the trial-specific precision-weighted PE of advice volatility (*Equation 13*);
11. Volatility Card Precision-weighted PE – parametric modulator of (5), using the trial-specific precision-weighted PE of card color volatility (see *Equation 15*).

We observed no significant correlations between response times (RTs) and any of the parametric modulators ($|r| < 0.3$, $p > 0.05$) and therefore did not model RT explicitly. The lack of effects on RTs may be due to the temporal structure of our task (*Figure 1*). Specifically, participants responded long after having received individual information (card outcome in previous trial) and social information had fixed duration (video). Therefore, they are likely to have simply conveyed the decision in the response phase but made it at some time during the video or even before.

Second, we predicted that arbitration should be sensitive to volatility, and favor one or the other source of information as a function of perceived relative reliability. Based on this hypothesis, we also performed a non-model based, factorial analysis by dividing the 160 trials into four conditions corresponding to those factors (*Figure 10a*). This GLM included for each of the four conditions the time when the advice was presented (the social information phase) and the trial-wise wager amount as a parametric modulator. We assumed that the difference between the four conditions will be expressed in the advice phase, before participants make their predictions.

## Group level

Contrast images from the 38 participants entered a random effects group analysis (*Penny and Holmes, 2007*). We used F-tests to identify undirected arbitration signals. Moreover, one-sample t-tests to investigate directed social or individual arbitration signals and positive or negative BOLD responses for each of the computational trajectories of interest described above.

Participant gender and age were included as covariates of no interest at the group level (the findings remained the same without these covariates). To investigate individual variability in the representation of social arbitration as a function of reliance on advice, we used parameter $\zeta$ to perform a median split of the group of participants.

For all analyses, we report results that survived whole-brain family-wise error (FWE) correction at the cluster level at $p<0.05$, under a cluster-defining threshold of $p<0.001$ at the voxel level using Gaussian random field theory (*Worsley et al., 1996*). Given recent debate regarding the vulnerabilities of cluster-level FWE procedures (*Eklund et al., 2016*), it is worth emphasising that this cluster-defining threshold ensures adequate control of cluster-level FWE rates in SPM (*Flandin and Friston, 2016*). The coordinates of all brain regions were expressed in Montreal Neurological Institute (MNI) space.

Based on recent results that precisions at different levels of a computational hierarchy may be encoded by distinct neuromodulatory systems (*Payzan-LeNestour et al., 2013*; *Schwartenbeck et al., 2015*), we also performed ROI analyses based on anatomical masks. We included (i) the dopaminergic midbrain nuclei substantia nigra (SN) and ventral tegmental area (VTA) using an anatomical atlas based on magnetization transfer weighted structural MR images (*Bunzeck and Düzel, 2006*), (ii) the cholinergic nuclei in the basal forebrain and the tegmentum of the brainstem using the anatomical toolbox in SPM12 with anatomical landmarks from the literature (*Naidich and Duvernoy, 2009*) and (iii) the noradrenergic locus coeruleus based on a probabilistic map (*Keren et al., 2009*) (see *Figure 8—figure supplement 1* for this neuromodulatory ROI).

## Code availability

The routines for all analyses are available as Matlab code: https://github.com/andreeadiaconescu/arbitration (*Kasper and Diaconescu, 2020*; copy archived at https://github.com/elifesciences-publications/arbitration). The instructions for running the code in order to reproduce the results can be found in the ReadMe file.

## Acknowledgements

We are grateful for support by the Swiss National Science Foundation (Ambizione grant PZ00P3_167952 to AOD; PP00P1_150739, 100014_165884, and 100019_176016 to PNT) and the Krembil Foundation to AOD. We are also grateful to Klaas Enno Stephan for providing guidance and funding for the study.

## Additional information

### Funding

| Funder | Grant reference number | Author |
| --- | --- | --- |
| Swiss National Foundation | PZ00P3_167952 | Andreea Oliviana Diaconescu |
| Swiss National Foundation | PP00P1_150739 | Philippe N Tobler |
| Swiss National Foundation | 100014_165884 | Philippe N Tobler |
| Swiss National Foundation | 100019_176016 | Philippe N Tobler |
| Krembil Foundation | | Andreea Oliviana Diaconescu |

The funders had no role in study design, data collection and interpretation, or the decision to submit the work for publication.

## Author contributions
Andreea Oliviana Diaconescu, Conceptualization, Resources, Data curation, Software, Formal analysis, Supervision, Validation, Investigation, Visualization, Methodology, Writing - original draft, Project administration, Writing - review and editing; Madeline Stecy, Conceptualization, Data curation, Formal analysis, Investigation, Writing - original draft, Project administration, Writing - review and editing, Data acquisition; Lars Kasper, Conceptualization, Software, Formal analysis, Methodology, Writing - original draft, Writing - review and editing; Christopher J Burke, Conceptualization, Methodology, Writing - review and editing; Zoltan Nagy, Software, Methodology, Writing - review and editing; Christoph Mathys, Conceptualization, Software, Formal analysis, Validation, Methodology, Writing - review and editing; Philippe N Tobler, Conceptualization, Resources, Formal analysis, Supervision, Funding acquisition, Writing - original draft, Project administration, Writing - review and editing

## Author ORCIDs
Andreea Oliviana Diaconescu (iD) https://orcid.org/0000-0002-3633-9757
Lars Kasper (iD) http://orcid.org/0000-0001-7667-603X

## Ethics
Human subjects: Informed consent, and consent to publish, was obtained from all participants. The study was approved by the Ethics Committee of the Canton of Zürich (KEK-ZH 2010-0327). All participants gave written informed consent before taking part in the study.

## Decision letter and Author response
Decision letter https://doi.org/10.7554/eLife.54051.sa1
Author response https://doi.org/10.7554/eLife.54051.sa2

# Additional files

## Supplementary files
• Supplementary file 1. Main effects of precision-weighted outcome prediction errors. MNI coordinates and F-statistic of activations induced by precision-weighted prediction error about individually estimated card color probability (*Equation 14*). Related to *Figure 6—figure supplement 1a*. (B) MNI coordinates and F-statistic of activations induced by precision-weighted prediction error about advice validity (*Equation 8*). Related to *Figure 6—figure supplement 1b*.

• Transparent reporting form

## Data availability
Data generated during this study are available in Dryad under the doi:10.5061/dryad.wwpzgmsgs. Source data files have been provided for the main tables and figures. The routines for all analyses are available as Matlab code: https://github.com/andreeadiaconescu/arbitration (copy archived at https://github.com/elifesciences-publications/arbitration). The instructions for running the code in order to reproduce the results can be found in the ReadMe file.

The following dataset was generated:

| Author(s) | Year | Dataset title | Dataset URL | Database and Identifier |
|---|---|---|---|---|
| Diaconescu AO, Stecy M, Kasper L, Burke CJ, Nagy Z, Mathys C, Tobler PN | 2020 | Neural Arbitration between Social and Individual Learning Systems | https://doi.org/10.5061/dryad.wwpzgmsgs | Dryad Digital Repository, 10.5061/dryad.wwpzgmsgs |

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
