## [Decision Letter]

Thank you for submitting your article "Neural Arbitration between Social and Individual Learning Systems" for consideration by *eLife*. Your article has been reviewed by three peer reviewers, one of whom is a member of our Board of Reviewing Editors, and the evaluation has been overseen by Michael Frank as the Senior Editor. The following individual involved in review of your submission has agreed to reveal their identity: Jan Gläscher (Reviewer #2).

The reviewers have discussed the reviews with one another and the Reviewing Editor has drafted this decision to help you prepare a revised submission.

While the reviewers find the topic interesting and think your paper is well-written, they raised several major concerns regarding the contribution of the work and the interpretation of the findings.

I will not repeat the reviewer's other comments here, but will highlight some of them.

There is a major concern about the novelty of the work given that the task used in this work is just a small modification of a task used many times before (reviewer #3). Reviewers #1 and #3 also question whether the task actually measures the 'arbitration' between social and individual information. Relatedly, the reviewers think it remains unclear if the participants truly believed the social information was coming from other people. I also suggest authors further discuss the neural findings in the context of social vs. non-social information. Lastly, reviewer #3 questions if the wager truly reflects arbitration between multiple information.

Reviewer #1:

Diaconescu and colleagues examined the computational and neural correlates of arbitration between self-gathered information and advice from others ('social' information). To enable a factorial analysis of information source and volatility, authors used 2x2 design (low vs. high volatility phases x two sources of information). Thirty eight individuals participated in a probabilistic learning task where they predicted the outcome of a lottery and hierarchical gaussian filter (HGF) was used to model the choice behavior. Behaviorally, authors found that volatility affected choice accuracy and amount of points wagered, which was consistent with existing literature. Model-based fMRI results showed that arbitration based on self-gathered information activated the midbrain and DLPFC whereas arbitration based on advice from others activated the amygdala and the vmPFC.

This is an elegant application of HGF to investigate the arbitration between multiple sources of information. I think the paper is well-written and authors rigorously compared multiple computational models and overall their methodology is strong.

1) As the authors also acknowledged in the subsection “Conclusions”, it is unclear if authors could examine the arbitration between non-social vs. 'social' information. In Diaconescu et al., 2014, from which authors adopted and modified a task, a pair of participants were invited to study decision-making in social interaction but in this study, it was not like that and I'm not sure if we can call it 'social information'. So, the findings reported in this work might be just related to arbitration between self-gathered information and self-perception of reliability of another source of information, which hampers my enthusiasm about this work.

2) Related to the previous comment, it would be useful to know how many participants actually believed they are playing with a human advisor. Also, please provide the instructions given to participants (e.g. were they told under what circumstances advisor is incentivized to give wrong/correct advice?)

Reviewer #2:

The authors describe a study that utilizes a variant of the "Advisor task", which was presented in a previous publications (PLoS CB, 2014 and SCAN, 2017) and which involved the binary decision for one of two lotteries in the presence of social advice. In this variant the author introduce a wager on the decision, which is affected by the volatility and the source of the information (card, i.e. own experience vs. advisor, i.e. social information). In a model-free analysis they show that the these two factors affect the decision, the advice-taking behavior and the wager that they place on their decision, which directly influence the trial-by-trial payoffs. The general finding was that decision accuracy was better during stable reward contingencies, whereas the same effect was more pronounced for advice-taking and wager size for the social information from the advisor. Using Hierarchical Gaussian Filters (HGF), the authors report that a fully hierarchical Level-3 HGF provide the best fit to the data. Several of the model's internal variables (amongst others the belief uncertainty, the arbitration between the different sources of information) were submitted to a model-based fMRI analysis, which identified a wide-spear network of brain regions that correlated with the arbitration signal. Further analyses suggested that arbitration in favor of the own experience correlated with activity in the amygdala and OFC, whereas arbitration in favor of the social information correlated activity in substantia nigra, DLPFC, insula and occipito-temporal and inferior temporal cortex. A separate ROI analysis constrained to neuromodulatory nuclei in the mid-brain also showed correlations with the arbitration signal.

The paper employs a task that is well-suited for dissociating the influence of different sources of information and renders itself well for computational modeling using the HGF. The findings are timely given recent report on the arbitration of model-free and model-based RL in the ventrolateral PFC. The paper is well-written and should if interest to the wide and sophisticated readership of *eLife*.

I only have a few suggestions for improvements of the manuscript:

1) Upon my first read-through, I found myself wondering what "prediction accuracy" (subsection “Behaviour: Prediction accuracy and wager size”) was referring to. In Figure 1, the choice of the subject is framed as a "decision", and it is only in the legend that this is referred to as prediction of a lottery. I think it would help the read to straighten out the terminology in the task description.

2) The BOLD time courses in Figure 7B look strange as they show the inverted shape from the normal BOLD response. Can the authors explain what is going on here?

3) The swoosh as the color bar is mostly meaningless in all the figures as one can only see the thresholded maximum value in the SPMs. I suggest to remove them (though I admit that they look cool).

4) The ROI analysis of mid-brain neuromodulatory nuclei needs to be better justified. The analysis pops up almost out of nowhere. It is clearly a relevant finding, but it should be stated more explicitly, why arbitration signals in these mid-brain nuclei are relevant for the current research question.

Reviewer #3:

Diaconescu et al. use a small modification of a previous task used many times before (Diaconescu et al., 2014, 2017; Behrens et al., 2008; Cook et al., 2019, to name a few studies) to examine the arbitration between individual and social advice learning. They test a good sample size of participants, and the addition of a trial by trial wager is interesting. However, I feel with the paradigm has been used so many times before that the study does not tell us anything particularly new. There is also a lot of visual activation in the individual learning condition and the Introduction and Discussion seem a bit disjointed. The fMRI results are also not particularly anatomically motivated, and just read like a long list of brain areas.

Does a model that was able to capture behaviour in the original task the authors used, with a dynamic learning rate (Behrens et al., 2007; 2008) perform worse than the behaviour estimated by the HGF? Moreover, there is an increasing appreciation that model comparison should not be the only way to decide between different models, but the parameters from the winning model should also be recoverable (Palminteri et al., 2017). Are the different model parameters recoverable?

In the Introduction the authors only discuss a putative role in the task for the dlPFC, TPJ and dmPFC, but very similar versions of the task have shown other areas to be involved, such as ventral striatum and different portions of the cingulate cortex. I feel the predictions about potential brain areas should relate more closely to the previous literature.

What are the correlations between the different time periods and parametric modulators in the GLM?

The authors justify not having a non-social control, but it is very difficult to interpret the results as they are not subtracted from another matched condition in the main analysis. This seems to be a general problem with the task itself that makes it very difficult to dissociate self and other relevant information. Indeed, studies by Cook et al. suggest a key difference between the social and non-social components in the task is that the social component represents an additional source of information to learn about, so is not just different in the social vs. non-social nature.

I am not convinced that this task measures the 'arbitration' between social and individual information. The authors state that the number of points wagered reflects 'arbitration' but does this measure not reflect confidence in the judgement? Also, as participants are not making separate wagers about the reliability of the reward and social information it is hard to know what precisely is influencing their decision.

How do the authors know that the participants believed the social information was from real other people?

---

## [Author Response]

While the reviewers find the topic interesting and think your paper is well-written, they raised several major concerns regarding the contribution of the work and the interpretation of the findings.I will not repeat the reviewer's other comments here, but will highlight some of them.There is a major concern about the novelty of the work given that the task used in this work is just a small modification of a task used many times before (reviewer #3). Reviewers #1 and #3 also question whether the task actually measures the 'arbitration' between social and individual information. Relatedly, the reviewers think it remains unclear if the participants truly believed the social information was coming from other people. I also suggest authors further discuss the neural findings in the context of social vs. non-social information. Lastly, reviewer #3 questions if the wager truly reflects arbitration between multiple information.

Thank you very much for sending the paper to experts in the field. Your and their suggestions helped us greatly improve the manuscript. We now describe that by carefully varying the relative validity of the sources of information and comparing models with arbitration against models without arbitration, our study supports the notion that arbitration is an important aspect of decisions that require integration of multiple sources of information. Following your suggestion, we now discuss the neural findings in the context of social vs. non-social representations:

“With respect to social vs. non-social learning signatures, we observed that the sulcus of the ACC represents predictions related to one’s own estimates of the card color outcomes, whereas the subgenual ACC represents predictions about the advisor’s fidelity. This is consistent with previous findings that the sulcus of the ACC dorsal to the gyrus plays a domain-general role in motivation (Apps et al., 2016; Rushworth and Behrens, 2008; Rushworth et al., 2007), whereas the gyrus of the ACC signals information related to other people (Apps et al., 2013, 2016; Behrens et al., 2008; Lockwood, 2016).”

Moreover, we now include additional behavioural and debriefing results that further clarify what participants believed and how they used the social advice when performing the task. We also more carefully describe that we treated wager magnitude as a measure of decision confidence. Indeed, before our study, it remained an open question whether confidence varies as a function of arbitration, in addition to precision/uncertainty of estimated information. Thus, our study provides novel insights also for researcher interested in confidence. In sum, addressing all the comments of the reviewers has greatly helped us to improve our manuscript.

Reviewer #1:[…]1) As the authors also acknowledged in the subsection “Conclusions”, it is unclear if authors could examine the arbitration between non-social vs. 'social' information. In Diaconescu et al., 2014, from which authors adopted and modified a task, a pair of participants were invited to study decision-making in social interaction but in this study, it was not like that and I'm not sure if we can call it 'social information'. So, the findings reported in this work might be just related to arbitration between self-gathered information and self-perception of reliability of another source of information, which hampers my enthusiasm about this work.

Thank you for raising this issue. As is often the case, our study needed to address the tension between ecological validity and experimental control. When we first developed the paradigm, the advisors were actual participants performing the binary lottery task in real-time (cf. Diaconescu et al., 2014). Previous studies using similar versions of this paradigm (see Cook et al., 2019; Diaconescu et al., 2014; Sevgi et al., 2020) have found that the order of congruent and incongruent advice phases has an impact not only on participants’ performance and degree of adherence to the advice, but also on the model parameter estimates. This is unsurprising, since the parameter estimates depend on both the inputs and responses. This is why we decided to adapt the paradigm of Diaconescu et al., 2014, to ensure that advice validity was constant across participants.

To standardize the advice for the presented study, we videotaped the decisions of the advisors, instructed them to display as little emotion as possible, and selected only videos of advisors who utilized the dominant strategy, i.e., advising participants according to the incentive structure they received prior to the start of the experiment. Note that it is standard practice in behavioural economic studies investigating social exchange to use previously recorded behavior in order to increase experimental control (Crockett et al., 2017; Dreher et al., 2016; Engelmann et al., 2019). To increase ecological validity compared to this body of research, we took advantage of the inherently social nature of actual humans giving advice in videos showing their faces and hands. We now explain the rationale for our choice of design better (Materials and methods).

“To compare participants in terms of their learning and decision making parameters, we needed to standardize the advice. This means that each participant received the same input sequence, i.e., order and type of videos.”

“To standardize the advice, avoid implicit cues of deception, and make the task as close as possible to a social exchange in real time, the videos of the advisor were extracted from trials when they truly intended to help or truly intended to mislead. Although each participant received the same advice sequence throughout the task, the advisors displayed in the videos varied between participants, in order to ensure that physical appearance and gender did not impact on their decisions to take advice into account.”

To investigate the social validity of our task, we now use the *debriefing questionnaire* answers as well as the *participants’ ratings of the advice* during the task. These data suggest that the majority of participants experienced the advice as intentional, in line with the belief that advice information indeed had a social origin.

“Debriefing Questionnaire

After completing the task, participants filled out a task-specific debriefing questionnaire, assessing their perception of the advisor and how they integrated the social information during the task. […] Thus, participants experienced advisors as intentional and helpful, which are core characteristics of social agents.”

“We used classical multiple regression and post-hoc tests to examine whether the model parameter estimates extracted from the winning model (M_1_) explained participants’ advisor ratings, as measured by debriefing questions after the main experiment outside the scanner. […] Thus, not only perceived participants the advice in our task as intentional and helpful, our model also explained some of these impressions.”

It is also worth noting that in a previous study (Diaconescu et al., 2014), we have characterized the best-fitting models for participants who face less social (i.e., nonintentional) advice from blindfolded advisors. According to these models, participants did not incorporate time-varying estimates of volatility about the advisor into their decisions. Importantly, in the current study, models without volatility performed substantially worse than hierarchical models (see Figure 2 and Table 2A for details). Thus, our participants appeared to process advisor intentionality, in-keeping with the notion that they indeed processed advice as social in nature. We describe this as follows:

“In order to distinguish general inference processes under volatility from inference specific to intentionality, we previously included a control task (Diaconescu et al., 2014), in which the advisor was blindfolded and provided advice with cards from predefined decks that were probabilistically congruent to the actual card colour. […] In the current study, we tested this by including models without volatility, but found that they performed substantially worse than hierarchical models (see Figure 2 and Table 2A for details). Thus, our participants appeared to process advisor intentionality.”

Two additional aspects of the data indicate that the participants processed social information quite differently from individual information, contrary to what one would expect if they simply integrated two variants of self-perceived information. First, advisor ratings during the task suggest that participants’ impressions of the advisor were more strongly influenced by advice than card volatility:

“Advisor Ratings

Participants were asked to rate the advisor (i.e., helpful, misleading, or neutral with regard to suggesting the correct outcome) in a multiple choice question presented 5 times during the experiment. […] This suggests that advice ratings decreased during volatile compared to stable phases, and this effect was more strongly related to the advice compared to the card cue.”

Second, thanks to the reviewer’s comment, we noticed an error in the manuscript, with respect to the analysis of the social weighting parameter zeta. In brief, we had tested whether the social weighting parameter significantly differed from 1, the prior for zeta which reflected equal weighting of the social and non-social cues, in order to examine whether participants preferentially weighted the advice over their own estimates of the card colours. However, the prior value of zeta is equivalent to log(1), i.e., 0 and not 1. Correcting this error revealed that social weighting was significantly above log(1), suggesting that participants preferentially relied on the advice to learn about the task outcome. See Results section for the correction in the manuscript.

“The reliability-independent social bias parameter ζ differed significantly from zero (t(36) = 5.09, p = 1.07e-05). […] Thus, on average, participants relied more on the advisor’s recommendations compared to their own sampling of the card outcomes (Figure 4C).”

2) Related to the previous comment, it would be useful to know how many participants actually believed they are playing with a human advisor.

As described in the response to the previous question, the debriefing question # 3 “Did the advisor intentionally use a strategy during the task? If yes, describe what strategy that was” suggests that 30 out of 38 participants believed their advisor used a strategy and intentionally helped or misled them at different phases during the task. When we asked the remaining 8 participants why they answered “No” to this question, they reported that they thought the advisor was using a random strategy. We repeated the analyses including only the 30 participants who perceived the advisor as intentionally trying to help or mislead at various times during the task, and found that all conclusions remained statistically the same.

Also, please provide the instructions given to participants (e.g. were they told under what circumstances advisor is incentivized to give wrong / correct advice?)

We now include what participants were told about how advisors were incentivized in the Materials and methods section.

The task instructions and debriefing questionnaire, which were originally presented to participants in their native German, were translated into English for the purpose of this paper. Pronouns were adapted to the advisor’s gender:

Task instructions

“The advisor has generally more information than you about the outcome on each trial. […] Nevertheless, he/she will on average have better information than you and his/her advice may be valuable to you.”

Reviewer #2:[…] I only have a few suggestions for improvements of the manuscript:1) Upon my first read-through, I found myself wondering what "prediction accuracy" (subsection “Behaviour: Prediction accuracy and wager size”) was referring to. In Figure 1, the choice of the subject is framed as a "decision", and it is only in the legend that this is referred to as prediction of a lottery. I think it would help the read to straighten out the terminology in the task description.

Our apologies for the confusion, we have now corrected the terminology in Figure 1 (replacing “Decision” with “Prediction”). Moreover, we describe the dependent measure as “Accuracy of lottery outcome prediction” in the Behavioural Results section and refer to the prediction of the *lottery outcome* throughout the manuscript.

2) The BOLD time courses in Figure 7B look strange as they show the inverted shape from the normal BOLD response. Can the authors explain what is going on here?

Thank you very much for raising this issue, which helped us realize that there was a sign error in the plotting of one of the ROI timeseries in the previous version of the figure.

We reran all the analyses for the revision (now also including the trial number index as an additional parametric modulator to control for fatigue). We also reran the BOLD time series extraction. Most extracted BOLD time series have a typical shape, with increasing BOLD signal intensity following trial onset (note that it is not uncommon for ventral prefrontal regions to show initial BOLD signal decreases).

Accordingly, we have updated Figure 7B.

3) The swoosh as the color bar is mostly meaningless in all the figures as one can only see the thresholded maximum value in the SPMs. I suggest to remove them (though I admit that they look cool).

Thank you for your suggestion. We have now removed them throughout the figures.

4) The ROI analysis of mid-brain neuromodulatory nuclei needs to be better justified. The analysis pops up almost out of nowhere. It is clearly a relevant finding, but it should be stated more explicitly, why arbitration signals in these mid-brain nuclei are relevant for the current research question.

Thank you for pointing this out. We have added the following paragraph to the Introduction section of the paper.

“An additional intriguing question is which neuromodulatory system supports the arbitration process. […] Here, we examined the unique contribution of arbitration to activity across dopaminergic, cholinergic, and noradrenergic neuromodulatory systems.”

We also refer to this in the Materials and methods section:

“Based on recent results that precisions at different levels of a computational hierarchy may be encoded by distinct neuromodulatory systems (Payzan-LeNestour et al., 2013; Schwartenbeck et al., 2015), we also performed ROI analyses based on anatomical masks. We included (i) the dopaminergic midbrain nuclei substantia nigra (SN) and ventral tegmental area (VTA) using an anatomical atlas based on magnetization transfer weighted structural MR images (Bunzeck and Düzel, 2006), (ii) the cholinergic nuclei in the basal forebrain and the tegmentum of the brainstem using the anatomical toolbox in SPM12 with anatomical landmarks from the literature (Naidich and Duvernoy, 2009) and (iii) the noradrenergic locus coeruleus based on a probabilistic map (Keren et al., 2009) (see Figure 8—figure supplement 1 for this neuromodulatory ROI).”

Reviewer #3:Diaconescu et al. use a small modification of a previous task used many times before (Diaconescu et al., 2014, 2017; Behrens et al., 2008; Cook et al., 2019, to name a few studies) to examine the arbitration between individual and social advice learning. They test a good sample size of participants, and the addition of a trial by trial wager is interesting. However, I feel with the paradigm has been used so many times before that the study does not tell us anything particularly new. There is also a lot of visual activation in the individual learning condition and the Introduction and Discussion seem a bit disjointed. The fMRI results are also not particularly anatomically motivated, and just read like a long list of brain areas.

Thank you for your feedback regarding the paradigm and the presentation of the fMRI results. We have revised the Introduction and the Discussion section to provide a more cohesive overview (see specific responses below).

Regarding the paradigm, we would like to point out that although the introduction of the wager is a relatively small modification to this type of task, it makes two important contributions. First, it provides a behavioural expression of decision confidence in terms of the number of points one is willing to win or lose based on a decision that has already been made. Secondly, it allows us to capture not only the binary decision as behavioural readout from each trial but provides a more continuous measure. This facilitates the estimation of a large number of parameters pertaining to both the perceptual and response models – see “Parameter recovery” subsection for details. This is because two sets of responses enter the computation of the posterior *maximum* a posteriori estimates as well as the model evidence and because sensitivity to model parameter changes is typically higher in data with continuous readout variables than categorical ones.

Does a model that was able to capture behaviour in the original task the authors used, with a dynamic learning rate (Behrens et al., 2007; 2008) perform worse than the behaviour estimated by the HGF?

Thank you for this question. We have now included a set of normative models to address this question. The answer to the question indeed is that the winning model identified in the present study explains behaviour better than a normative model of learning with a dynamic learning rate (cf. Behrens et al., 2007, 2008). While the model used by Behrens et al., 2007, 2008 assumes a normative learning process, the winning model in this study is one where the perceptual model parameters (i.e., parameters capturing learning from advice and card colour outcomes) are estimated from participants’ responses.

In the revision of the paper, we included a normative model family as an alternative candidate in our model space, and included perceptual parameter estimates that were fixed to their prior values (see Table 1). This assumes optimal dynamic Bayesian learning across participants. The only parameters estimated for this normative model family are the response model parameters. For model comparison, we used the log model evidence (LME), which trades off model complexity for accuracy. Thus also when accounting for model complexity, our non-normative 3-level HGF explained behaviour better than the other models (Results section).

“We used computational modelling with hierarchical Gaussian Filters (HGF; Figure 2) to explain participants’ responses on every trial. […] For model comparison, we used the log model evidence (LME), which represents a trade-off between model complexity and model fit.”

Results section:

“The winning model was the 3-level HGF with arbitration (*ϕ_p_*= 0.999; Bayes Omnibus Risk = 4.26e-11; Figure 3B; Table 2A). […] The model family that included volatility of both information sources outperformed models without volatility, in-keeping with the model-independent finding that perceived volatility of both information sources affected behaviour.”

Moreover, there is an increasing appreciation that model comparison should not be the only way to decide between different models, but the parameters from the winning model should also be recoverable (Palminteri et al., 2017). Are the different model parameters recoverable?

Thank you for this suggestion. We completely agree and have now included a section on parameter recovery and new Figure 2—figure supplement 1.

“The winning 3-level full HGF model includes multiple parameters that need to be estimated. […] Based on this criterion, we could recover all parameters well, as all Cohen’s *f* values equaled or exceeded 0.4 (see Figure 2—figure supplement 1).”

In the Introduction the authors only discuss a putative role in the task for the dlPFC, TPJ and dmPFC, but very similar versions of the task have shown other areas to be involved, such as ventral striatum and different portions of the cingulate cortex. I feel the predictions about potential brain areas should relate more closely to the previous literature.

Thank you for the suggestion. We have now adjusted the Introduction accordingly.

“It is also worth noting that arbitration depends on both experienced and inferred value learning. […] In addition to being associated with volatility tracking in a probabilistic reward learning task (Behrens et al., 2007), the ACC was shown to represent volatility precision-weighted PEs during social learning (Diaconescu et al., 2017).”

What are the correlations between the different time periods and parametric modulators in the GLM?

Overall, the pairwise correlations between the different time periods and the parametric modulators were small, as can be seen from the averaged correlation matrix computed across all participants (Figure 1—figure supplement 2). The largest correlations arose between the two sets of hierarchical precision-weighted prediction errors (PEs) about the card colour outcome. The lower-level precision-weighted PEs reflect Bayesian surprise, the absolute value of the difference between the outcome and the expected card colour probability. We did not orthogonalise any of the regressors because we were interested in the unique variance explained by each regressor in our design matrix. A strong correlation would lead to reduced sensitivity for detecting unique effects. In our case, the effects in question were the neural representations of the hierarchical precision-weighted card colour PE (see new Figure 1—figure supplement 2). Our analysis of these effects revealed similar effects as those described previously (Figure 7—figure supplements 2-3), in line with sensitivity being comparable to that of previous research.

The authors justify not having a non-social control, but it is very difficult to interpret the results as they are not subtracted from another matched condition in the main analysis. This seems to be a general problem with the task itself that makes it very difficult to dissociate self and other relevant information. Indeed, studies by Cook et al. suggest a key difference between the social and non-social components in the task is that the social component represents an additional source of information to learn about, so is not just different in the social vs. non-social nature.

Thank you for raising this point. It is important to note that the investigation of the arbitration process is independent of the social vs. non-social distinction. We agree that the social component of the task included advice, i.e., an additional source of information that participants could, and did, learn about. Like card probability, it is imbued with uncertainty, since participants do not know how much more insight the advisor has about the outcome of the lottery. In this respect, and with regard to the fact that they both occur in every trial, are associated with similar reward probabilities and their volatility varies independently and in a block-wise fashion, the two sources of information are matched. Whether the advice is specifically social in nature or rather leads to general learning from an indirect and uncertain information source could be examined in more detail by including an additional control condition. We now highlight this in the study Limitations section:.

“Second, we did not include a non-social control task. […] However, whether the process we identified is specifically social in nature or rather reflects learning from an indirect information source needs to be examined in future studies by including an additional control condition.”

We now include additional analyses and comparisons with a previous study that speak to this issue. First, we now examine the *debriefing questionnaire* answers as well as the *participants’ ratings of the advice* during the task. These data suggest that the majority of participants experienced the advice as intentional, in line with the belief that advice information indeed had a social origin (in line with our use of videos of actual people raising cards with a particular colour).

“Debriefing Questionnaire

After completing the task, participants filled out a task-specific debriefing questionnaire, assessing their perception of the advisor and how they integrated the social information during the task. […] Thus, participants experienced advisors as intentional and helpful, which are core characteristics of social agents.”

“We used classical multiple regression and post-hoc tests to examine whether the model parameter estimates extracted from the winning model (M_1_) explained participants’ advisor ratings, as measured by debriefing questions after the main experiment outside the scanner. […] Thus, not only perceived participants the advice in our task as intentional and helpful, our model also explained some of these impressions.”

Second, in a previous study (Diaconescu et al., 2014), we have characterized the best fitting models for participants who face less social (i.e., non-intentional) advice from blindfolded advisors. According to these models, participants did not incorporate time-varying estimates of volatility about the advisor into their decisions. Importantly, in the current study, models without volatility performed substantially worse than models with volatility (see Figure 2 and Table 2A for details). Thus, our participants appeared to process advisor intentionality, in-keeping with the notion that they indeed processed advice as social in nature. We describe this as follows:

“In order to distinguish general inference processes under volatility from inference specific to intentionality, we previously included a control task (Diaconescu et al., 2014), in which the advisor was blindfolded and provided advice with cards from predefined decks that were probabilistically congruent to the actual card colour. […] Thus, our participants appeared to process advisor intentionality.”

Two additional aspects of the data indicate that the participants processed social information quite differently from individual information, contrary to what one would expect if they simply integrated two variants of self-perceived information. First, advisor ratings during the task suggest that participants’ impressions of the advisor were more strongly influenced by advice than card volatility:

“Advisor Ratings

Participants were asked to rate the advisor (i.e., helpful, misleading, or neutral with regard to suggesting the correct outcome) in a multiple choice question presented 5 times during the experiment. […] This suggests that advice ratings decreased during volatile compared to stable phases, and this effect was more strongly related to the advice compared to the card cue.”

Third, we also contrasted social compared to non-social representations by including social- and card-weighting as additional parametric modulators in the design matrix. (as follows) and Figure 7—figure supplement 1:

“With respect to social vs. non-social learning signatures, we observed that the sulcus of the ACC represents predictions related to one’s own estimates of the card colour outcomes, whereas the subgenual ACC represents predictions about the advisor’s fidelity. This is consistent with previous findings that the sulcus of the ACC dorsal to the gyrus plays a domain-general role in motivation (Apps et al., 2016; Rushworth and Behrens, 2008; Rushworth et al., 2007), whereas the gyrus of the ACC signals information related to other people (Apps et al., 2013, 2016; Behrens et al., 2008; Lockwood, 2016).”

I am not convinced that this task measures the 'arbitration' between social and individual information. The authors state that the number of points wagered reflects 'arbitration' but does this measure not reflect confidence in the judgement?

Thank you for raising this point. While the task does not measure arbitration directly, our model allows us to infer on the process. As suggested by the reviewer, the number of points wagered indeed provided us with a behavioural readout of decision confidence. We envisage confidence to reflect multiple factors and processes, one of them being arbitration. Our model defined arbitration in terms of hierarchical Bayesian inference, as the relative perceived reliability of each information source. In other words, arbitration was formalised as a ratio of precisions: the precision of the prediction about advice accuracy and colour probability, divided by the total precision (Materials and methods section). We clarified this further in the manuscript:

“The number of points wagered provided us with a behavioural readout of decision confidence. We aimed to formally explain trial-wise wager responses as a linear function of various sources of uncertainty and precision associated with the lottery outcome prediction: (i) irreducible decision uncertainty about the outcome, (ii) arbitration, (iii) informational uncertainty about the card colour or the advice, and (iv) environmental uncertainty/volatility about the card colour or the advice.”

Moreover, we now formulate more carefully in the Abstract:

“Decision confidence, as measured by the number of points participants wagered on their predictions, varied with our relative precision definition of arbitration.”

Also, as participants are not making separate wagers about the reliability of the reward and social information it is hard to know what precisely is influencing their decision.

Thank you for giving us the opportunity to clarify. While it is true that participants did not make separate wagers after each source of information, we tailored both the experimental design and the analysis to examine how each information source contributed to the trial-wise decisions. First, we manipulated volatility of each information source separately and used a factorial design, where trials could be divided into four conditions: (i) stable card and stable advisor, (ii) stable card and volatile advisor, (iii) volatile card and stable advisor, and (iv) volatile card and volatile advisor in a total of 160 trials. Our behavioural findings (accuracy of predicting lottery outcome, advice taking, points wagered, advisor ratings) indicate that participants process both sources of information. Secondly, modelling showed that both social and non-social aspects of uncertainty independently explained trial-wise wager magnitude in the response model (see Figure 5). Thirdly, we addressed the question of whether participants integrate the two sources of information or rather treat them separately by including different response model families. These included: (i) an “Arbitrated” model, which assumed that participants combine the two information sources, possibly unequally, (ii) an “Advice only” model, assuming arbitration-free reliance on social information only, and (iv) a “Card only” model, representing arbitration-free reliance on the inferred card colour probabilities only. Model selection results suggest that participants integrate the two sources of information to guide their decisions.

How do the authors know that the participants believed the social information was from real other people?

In addition to using videos that showed the faces and hands of people giving advice commensurate with actual intentions, we addressed this concern in three ways: First, the task instructions emphasised that the advisor had received privileged information about the lottery outcomes on every trial. Second, throughout the task, we asked participants to rate the fidelity of the advisor, and used those ratings to test the validity of the model predictions. Third, we also debriefed participants about their perception and reliance on the advisor. We now describe these measures in the Materials and methods section.

Participants were given the following instructions about the advisor:

“The advisor has generally more information than you about the outcome on each trial. […] Nevertheless, he/she will on average have better information than you and his/her advice may be valuable to you.”

Advisor ratings during the task also allowed us to capture participants’ impressions of the advisor (Results section).

“Advisor Ratings

Participants were asked to rate the advisor (i.e., helpful, misleading, or neutral with regard to suggesting the correct outcome) in a multiple choice question presented 5 times during the experiment. […] This suggests that advice ratings decreased during volatile compared to stable phases, and this effect was more strongly related to the advice compared to the card cue.”

Moreover, one of debriefing questions (“Did the advisor intentionally use a strategy during the task? If yes, describe what strategy that was”) directly measured participant beliefs. The responses suggest that 30 out of 38 participants believed their advisor used a strategy and intentionally helped or misled them at different phases during the task. When we asked the remaining 8 participants why they answered “No” to this question, they reported that they thought the advisor was using a random strategy. We repeated the analyses including only the 30 participants who perceived the advisor as intentionally trying to help or mislead at various times during the task, and found that all conclusions remained statistically the same.

“Debriefing Questionnaire

After completing the task, participants filled out a task-specific debriefing questionnaire, assessing their perception of the advisor and how they integrated the social information during the task. […] On average, participants reported that they followed the advice 60% of the time (mean ratings 60 ± 12), which significantly differed from chance (t(37) = 5.02, p = 1.29e-05).”

Moreover, it is worth noting that in a previous study (Diaconescu et al., 2014), we have characterized the best-fitting models for participants who face less social (nonintentional) advice from blindfolded advisors. According to these models, participants in these control situations did not incorporate time-varying estimates of volatility about the advisor into their decisions. Importantly, in the current study, models without volatility performed substantially worse than hierarchical models (see Figure 2 and Table 2A for details). Thus, our participants appeared to process advisor intentionality, in-keeping with the notion that they indeed processed advice as coming from real people. We describe this as follows:

“In order to distinguish general inference processes under volatility from inference specific to intentionality, we previously included a control task (Diaconescu et al., 2014), in which the advisor was blindfolded and provided advice with cards from predefined decks that were probabilistically congruent to the actual card colour. […] Thus, our participants appeared to process advisor intentionality.”

References:

Cook, J.L., Swart, J.C., Froböse, M.I., Diaconescu, A.O., Geurts, D.E., den Ouden, H.E., and Cools, R. (2019). Catecholaminergic modulation of meta-learning. *eLife* 8, e51439.

Crockett, M.J., Siegel, J.Z., Kurth-Nelson, Z., Dayan, P., and Dolan, R.J. (2017). Moral transgressions corrupt neural representations of value. Nat. Neurosci. 20, 879–885.

Dreher, J.-C., Dunne, S., Pazderska, A., Frodl, T., Nolan, J.J., and O’Doherty, J.P. (2016). Testosterone causes both prosocial and antisocial status-enhancing behaviors in human males. Proc. Natl. Acad. Sci. 113, 11633–11638.

Engelmann, J.B., Meyer, F., Ruff, C.C., and Fehr, E. (2019). The neural circuitry of affect induced distortions of trust. Sci. Adv. 5, eaau3413.

Sevgi, M., Diaconescu, A.O., Henco, L., Tittgemeyer, M., and Schilbach, L. (2020). Social Bayes: Using Bayesian Modeling to Study Autistic Trait–Related Differences in Social Cognition. Biol. Psychiatry 87, 185–193.